# Dopamine neurons encode trial-by-trial subjective reward value in an auction-like task

Daniel F. Hill [1] ✉, Robert W. Hickman[1], Alaa Al-Mohammad [1],
Arkadiusz Stasiak [1] & Wolfram Schultz [1] ✉

The dopamine reward prediction error signal is known to be subjective but has so far only been assessed in aggregate choices. However, personal choices fluctuate across trials and thus reflect the instantaneous subjective reward value. In the well-established Becker-DeGroot-Marschak (BDM) auction-like mechanism, participants are encouraged to place bids that accurately reveal their instantaneous subjective reward value; inaccurate bidding results in suboptimal reward ("incentive compatibility"). In our experiment, male rhesus monkeys became experienced over several years to place accurate BDM bids for juice rewards without specific external constraints. Their bids for physically identical rewards varied trial by trial and increased overall for larger rewards. In these highly experienced animals, responses of midbrain dopamine neurons followed the trial-by-trial variations of bids despite constant, explicitly predicted reward amounts. Inversely, dopamine responses were similar with similar bids for different physical reward amounts. Support Vector Regression demonstrated accurate prediction of the animals' bids by as few as twenty dopamine neurons. Thus, the phasic dopamine reward signal reflects instantaneous subjective reward value.

"Beauty is no quality in things themselves: it exists merely in the mind which contemplates them, and each mind perceives a different beauty" (David Hume, 1711–1776[1]).

The value of a reward is in the eye of the beholder: it is determined by the specific benefit the reward provides for the individual agent. Thus, reward value is fundamentally subjective. However, subjective reward value cannot be directly measured; it can only be inferred from an agent's choices.

Common behavioral methods infer subjective reward value from a series of choices. The underlying assumption is that each presented choice option elicits a subjective valuation process at the moment of its presentation or at the moment of the choice. Agents then behave "as if" they choose the option with the highest subjective value at this moment. In this way, researchers are able to use such choices for estimating the subjective value of an option. Given the requirement of multiple choices, the inferred value is an average from several trials.

Dopamine neurons are known to encode such subjective reward value, which is demonstrated by dopamine concentration changes with food and fluid satiety and hunger[2,3] and quantified by choice indifference points, temporal discounting functions, and economic utility functions[4–6]. As dictated by the behavioral estimation methods, these neurophysiological studies have averaged reward value from multiple choices. However, the subjective value of a reward relies on fundamentally stochastic brain processes and is therefore likely to vary from trial to trial, even for physically identical rewards.

Trial-by-trial assessments have traditionally been used for studying neuronal reward responses and are typical done in statistical regression analyses. Such studies have shown that dopamine reward prediction error responses at cell bodies and axons in monkeys and rodents vary with a number of important behavioral measures, including movement reaction time[7,8], learning of reward-predicting stimuli[9], expected reward time[10,11], decision confidence (subjectively

[1]Department of Physiology, Development and Neuroscience, University of Cambridge, Cambridge, UK. ✉e-mail: hill.danielf@gmail.com; ws42@pm.me

perceived reward probability; distinct from subjective probability weighting in economics)[12,13], and licking responses to probabilistic rewards predictors[14]. However, none of these studies tested how subjective reward value might reflect the intrinsic stochasticity varying from trial to trial despite constant objective reward amount.

To reveal instantaneous, trial-by-trial fluctuations of subjective reward value requires an elicitation mechanism in which participants can state their personal value on each choice. The Becker-DeGroot-Marschak second-price auction-like mechanism (BDM)[15] provides such an approach, which is also suitable for monkeys[16]. In the BDM task, the bidder states their own subjective value directly against a randomly bidding computer opponent on each trial. Behavioral validity checks demonstrate that inappropriate BDM bidding results in suboptimal outcomes; when bidding too low, the bidder risks losing the auction, and when bidding too high, the bidder risks paying too much. Thus, the BDM bid reflects, on a trial-by-trial basis, the true subjective value of the object at the moment of the bidding ("incentive compatibility").

Human studies have demonstrated the suitability of the BDM mechanism for assessing neural correlates of trial-by-trial fluctuations of subjective value of such distinct rewards as food items and movie trailers[17,18]. Hence, the BDM may also become a useful behavioral tool for assessing trial-by-trial fluctuations of subjective reward value with the high precision afforded by single-cell neurophysiological recording in animals.

The present study used the BDM mechanism to investigate trial-by-trial, instantaneous neuronal coding of subjective reward value. Before taking up neuronal recordings, both monkeys of this study underwent extensive behavioral training and experience over several years in various versions of a BDM task[16]. Being fully aware of the many pitfalls of short-term BDM performance in humans, the behavioral results demonstrated reliable bidding that closely approximated the subjective reward values estimated by conventional binary choices. We used the best-developed version of the BDM task that resulted in reliable performance and tested dopamine neurons whose reward prediction error signal is known to encode subjective reward value inferred from binary choices[6]. We found that dopamine signals followed trial-by-trial variations of BDM bids, and we found further that dopamine signals were similar for identical bids for different physical reward amounts. Thus, dopamine neurons encode instantaneous subjective reward values. Correspondingly, the dopamine signal decoded by a Support Vector Regression predicted BDM bids with high accuracy.

## Results

### Monkeys' bids reflect instantaneous subjective reward value
This study used two rhesus monkeys who had undergone extensive behavioral training and testing in tens of thousands of trials in several BDM implementations before they entered the neurophysiological study[16]. In the current investigation, the two animals bid for fixed volumes of juice reward against a computer opponent in the BDM task (Fig. 1a). The task contained a sequence of events (Fig. 1a, Supplementary Fig. 1), the most important of which being the onset of the three fractal stimuli that defined the respective three magnitudes of juice on offer. Following a task-initiation screen (trial start), monkeys were shown one of three fractal images representing three different juice volumes (Monkey V: 0.3 ml, 1.0 ml, and 1.7 ml; Monkey U: 0.2 ml, 0.45 ml, and 0.7 ml). After the fractal image, the animal was shown a "bid space" representing 1.2 ml water (Fig. 1a; hashed black and white fill). Forward and backward movement of a lever (right hand side) resulted in the upward and downward movement of the cursor that indicated the monkey's bid (represented by a magenta bar; Fig. 1a). Once the animal's bid stayed stable for 500 ms, the computer bid was shown (green bar; randomly sampled from a uniform distribution). If the monkey's bid was equal to or exceeded the computer bid (win), the animal received the corresponding juice reward and the remainder of

the water after subtracting a water amount that corresponded to the computer bid (indicated by reversed direction of the hash fill in Fig. 1a); thus, the green bar also served as cue for the water payout on win trials. However, if the monkey's bid was too high, the random computer bid may have also been too high, and, despite winning the auction, the animal would lose a proportional amount of water reward; this overpayment constituted a "cost" that prevented overbidding (Fig. 1b). By contrast, if the monkey's bid was lower than the computer bid, the animal lost the auction and received the full water budget (1.2 ml) but no juice. If the monkey's bid was too low, he would not receive juice; this "cost" prevented underbidding (Fig. 1b). Thus, by eliciting bids according to the price an agent is willing to pay, BDM reveals the agent's true subjective value ("incentive compatibility").

Three fractal cues for three reward magnitudes were trained extensively (>20,000 trials). The fractals for the respective juice magnitudes were displayed in pseudo-random order to generate prediction errors relative to the mean experienced reward magnitude. Monkeys' individual bids were consistently rank ordered, and their means correlated well with juice volume ($R^2 = 0.61$, $p < 0.05$; session average $R^2 = 0.46$, $p < 0.05$ in 96.9% of sessions, Spearman rank correlation) (Fig. 1c, d; $n = 227$ and $n = 309$ sessions for Monkey V and Monkey U, respectively). The bids fluctuated from trial-to-trial within experimental sessions (Fig. 1c) and from day-to-day between sessions (Supplementary Fig. 2a). Importantly, if these fluctuations were driven by changes in subjective value, bid variability over time should be consistent at all three reward levels. In other words, if the variability of bidding is systematic and based on subjective valuation, we would expect the change in subjective value to affect bidding in the same way for all three reward levels. We tested this possibility by analyzing the coherence of bids across reward levels. Coherence in this case can be described as how the bids for each reward magnitude "move together" over time. That is, coherence indicates whether changes in bidding for one reward magnitude reflect changes in bidding for another. We found that bid fluctuations were indeed coherent from trial-to-trial and from day-to-day (Supplementary Fig. 2b, c; see Supplementary Table 1 for summary statistics), suggesting that bid variability resulted from fluctuating changes in subjective value.

To identify the most critical variables influencing the bidding behavior, we fitted a Lasso regression model using 31 candidate variables (see "Methods") that might have affected the monkeys' bids. To avoid overfitting a regression with a maximum of possible variables, the Lasso regression eliminated regressors with low explanatory power (as defined by the lambda coefficient being one standard error above the mean squared error; Supplementary Fig. 2d). A total of nine regressors survived the Lasso elimination and seven were included in a mixed effects model (Eq. (1)) (two variables were analyzed separately to account for collinearity, see "Methods", Eq. (3)). We eliminated the influence of trial number and session number by including them only as random effects grouping variables for the intercept. The remaining factors that significantly affected the animal's bids included reward magnitude, starting bid, total liquid consumed, previous computer bid for the same reward magnitude, and previous bidding result (win/lose) for the same reward magnitude (Fig. 1e; adjusted $R^2_V = 0.50$, $R^2_U = 0.41$). Note that the use of only three reward magnitudes has reduced bidding nonlinearities reflecting the animal's risk attitude, as BDM bidding is inherently risky due to the randomly bidding computer opponent. To better understand how each regressor contributed to the monkeys' bids independently of reward magnitude, we eliminated the influence of reward magnitude by including it as a random effect grouping variable for the intercept. The resulting modified mixed effects model was used to examine how the following four key variables affected bidding: starting bid, total liquid consumed, previous computer bid for the same reward magnitude, and previous bidding result (win/lose) for the same reward magnitude (Eq. (2); reduced mixed effects model 1; Supplementary Fig. 2e). Due to collinearity among "win streak"/"lose

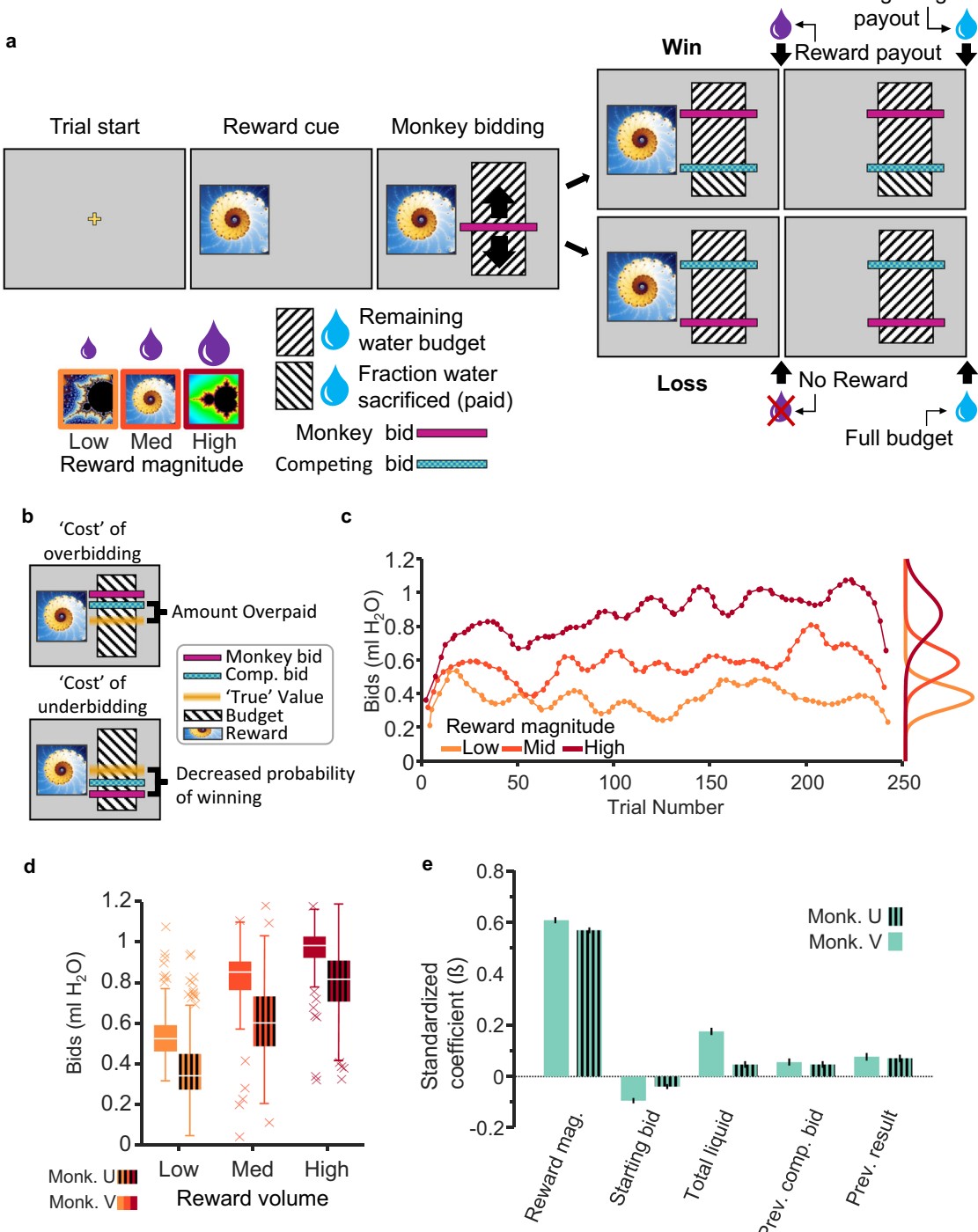

**Fig. 1 | Monkeys bid as if they are optimizing subjective reward value. a** Becker-DeGroot-Marschak auction-like bidding task. Monkeys were trained to associate fractal images with varying quantities of juice reward. Using a lever, they bid to indicate how much of their water endowment (refilled on each trial) they would be willing to "pay" for a given juice reward (willingness-to-pay; cyan bar). The "budget" (hashed area) was fixed at 1.2 ml of water. After bidding, the competing bid was displayed (cyan bar), and then rewards were paid out in sequence (i.e., juice delivery, 1.5 s delay, water delivery). **b** The optimal bidding strategy is to bid one's "true" value, which avoids overbidding (overpaying from the water endowment) and underbidding (less likely winning). **c** Example monkey bids from a single experimental session. Monkey bids were typically normally distributed and varied coherently within and between experimental sessions. **d** Average bids for all

sessions. Monkeys' bids were averaged for every reward level for each session. Box center lines represent median, margins represent 25th and 75th percentiles, whiskers represent the range of all non-outlier data, and "x" points indicate outliers ($n = 227$ and $n = 309$ sessions for monkey V and monkey U, respectively). Bids are rank-ordered on average and variance could largely be attributed to coherent changes in value over time (see Supplementary Fig. 2). **e** Mixed effects model illustrating predictors of monkey bidding (error bars show 95% confidence interval; $n = 18{,}604$ and $n = 24{,}528$ trials for monkeys V and U, respectively). Relevant task variables were identified using a lasso model (see Supplementary Fig. 2). A mixed-effects model was then used to determine their relative contributions to bidding independent of trial progression and between-day variability (see "Methods").

streak" and "previous result" variables, we performed a separate analysis to assess the role of multiple sequential wins or losses on bidding for the same reward magnitude by including "previous result" as a random effect and adding "previous win streak" and "previous lose streak", quantified by the number of sequential wins or losses for the same reward magnitude (Eq. (3); reduced mixed effects model 2; Supplementary Fig. 2f). Previous win/lose streak, starting bid, and total liquid consumed affected bidding differently for Monkey V than Monkey U: in monkey U, previous win streaks and previous lose streaks had greater influence over the animal's subsequent bid; in Monkey V, starting bid and liquid consumed were most influential (Monkey V: win streak $\beta = 0.016$, lose streak $\beta = -0.007$ ($p > 0.05$), starting bid $\beta = -0.1$ and total liquid $\beta = 0.18$; Monkey U: $\beta = 0.05$, $\beta = -0.03$, $\beta = -0.04$ and 0.47, respectively. $P < 0.05$ unless stated otherwise). Additionally, although the "previous result" and "win/lose streak" variables affected the animals' bids similarly (wins resulted in higher subsequent bids and losses in lower subsequent bids), consecutive wins and losses affected each animal to different degrees. These data suggest that different features of the task contributed uniquely to individual monkey's subjective value estimates. The systematic and consistent bidding over several months argues against distracted and random bidding behavior of these monkeys.

### Dopamine signal reflects trial-by-trial changes in subjective value

We recorded single-unit activity in the midbrain during performance of the BDM task. Neurons with wide waveforms (>1.8 ms) and low baseline impulse rates (<10 Hz) that responded significantly to at least one task event (analysis window: 0 to 200 ms post event; $p < 0.05$; Wilcoxon test) were categorized as putative dopamine neurons ($n = 145$ for Monkey V and $n = 123$ for Monkey U; $n =$ number of neurons); all other neurons were categorized as putative non-dopamine neurons ($n = 114$ for Monkey V and $n = 113$ for Monkey U). A number of dopamine neurons showed the typical two-component responses to the external reward cues documented and described before in much detail[5,6,19]. The initial, attentional response component consisted of an unspecific excitation elicited by the stimulus that likely reflected its detection, whereas the subsequent excitatory or inhibitory value response component reflected the bidirectional reward prediction error elicited by the reward-predicting stimulus or the primary reward. The relative composition of the two response components varied across the different bid levels, as described below, but only the second, value response component was likely to reflect the subjective value revealed by the bids.

Our neuronal data analysis focused on the three reward magnitude-predicting cues as the strongest and thus best quantifiable dopamine response in the BDM task (Fig. 1a). The three cues reflected higher-order reward prediction errors relative to the mean reward prediction from the immediately preceding trial-start cross (Fig. 1a). To analyze subjective value coding, we focused on the second, value component of dopamine response. By contrast, experience showed that the first, attentional dopamine response component was unrelated to value coding[5,6,19]. Indeed, it showed only inconsistent variations and was not further analyzed.

Roughly one-half to two-thirds of all dopamine neurons exhibited graded value responses (second response component) to the external reward cues (either the fractal indicating juice amount or the cue for water payout on win trials i.e., green bar; $n = 80$, 65 % for Monkey V and $n = 68$, 47 % for Monkey U; $p < 0.05$, two separate single linear regressions). Importantly, responses in a subset of these neurons correlated significantly with the monkeys' bids ($n = 41$ for Monkey V, and $n = 32$ for Monkey U; $p < 0.05$; single linear regression). The example neuron in Fig. 2a, b exhibited increased activity with increase of both reward magnitude and bid in response to onset of the fractal indicating the juice amount. Later analyses showed that the neuronal

response increased with bid independently of reward magnitude (see below).

We then selected dopamine neurons for significant correlation of their second, value response component with the bids, using the regression of Eq. (4) ($n = 41$ and $n = 32$ for monkeys V and U, respectively). These bid-encoding responses varied also with reward magnitude well (Fig. 2c, grayed area). The excitatory dopamine responses in trials with the fractal for the highest reward magnitude reflected the strong positive reward prediction error relative to the prediction by the immediately preceding stimulus, whereas the depression with the lowest reward magnitude reflected a negative reward prediction error; dopamine responses to the intermediate fractal consisted of a mild depression reflecting a mild negative reward prediction error. A similar quantitative pattern was seen when grouping the bid-encoding e-population responses of these bid-encoding neurons into quintiles of the bid space (Fig. 2d). By contrast, the first, attentional dopamine response component varied inconsistently with reward magnitude and bid level. The significant relationship of the bid-encoding dopamine neurons was also seen in both animals in the normalized population responses with reward magnitude (Fig. 2e; Monkey V: $p = 5.6 \times 10^{-15}$; Monkey U: $p = 3.5 \times 10^{-12}$; Kruskal–Wallis Test, Tukey–Kramer multiple comparison test) and with the bids (Fig. 2f; Monkey V: $R^2 = 0.88$, $p = 6.6 \times 10^{-12}$; Monkey U: $R^2 = 0.93$, $p = 4.1 \times 10^{-14}$; bids split into 25 bins; Eq. (4)). These relationships were also seen in all dopamine neurons (Supplementary Fig. 3). Neuronal population responses of the bid-encoding dopamine neurons varied significantly with bids for both monkeys. The same data for the population of all recorded dopamine neurons are shown in Supplementary Fig. 3. The average responses from bid quintiles are shown for individual bid-encoding dopamine neurons in Supplementary Fig. 4 ($n = 41$, Monkey V; $n = 32$, Monkey U) and for all dopamine neurons in Supplementary Fig. 5 ($n = 145$, Monkey V; $n = 123$, Monkey U).

The bidding required an arm movement and was therefore correlated with movement amplitude, velocity, and absement (movement amplitude x time). As dopamine responses are only very mildly modulated by movement compared to reward prediction error magnitude (Ljungberg et al. 1992; Satoh et al. 2003), our analysis focused on the bids. Indeed, testing for movement parameters in analogy to Eq. (4), we found only few dopamine neurons whose responses varied with movement velocity (1 and 10 of 123 and 145 dopamine neurons in Monkeys U and V, respectively), unsigned velocity (8 and 2 neurons), absement (2 and 5 neurons), or unsigned absement (3 and 5 neurons). Of the specifically bid-encoding dopamine neurons, even fewer neurons varied with movement velocity (only 0 and 3 of 32 and 41 neurons in Monkeys U and V, respectively), unsigned velocity (2 and 2 neurons), absement (0 and 1 neuron), or unsigned absement (1 and 2 neurons). All of these neuron numbers failed to exceed the 5% chance level. Thus, movement parameters failed to explain bid-encoding in dopamine neurons.

### Dopamine neurons reflect subjective value (bids) irrespective of reward magnitude

Graded coding of subjective value despite same reward magnitude. In this experiment, the bid reports the subjective value of the reward magnitude, and therefore the two variables are intercorrelated. Consequently, variability amongst bids for a given specific reward magnitude is solely contingent on changes in subjective value over time. Our aim was to understand whether and how dopamine neurons encode these subtle changes in subjective value. The response of the dopamine neuron shown in Fig. 3a, b varied significantly with the bids when only the fractal for the single middle reward magnitude was displayed ($p < 0.03$; single linear regression).

The normalized population responses with each of the three reward magnitudes showed monotonic bid coding despite constant reward magnitude indicated by the three fractals (our analysis window

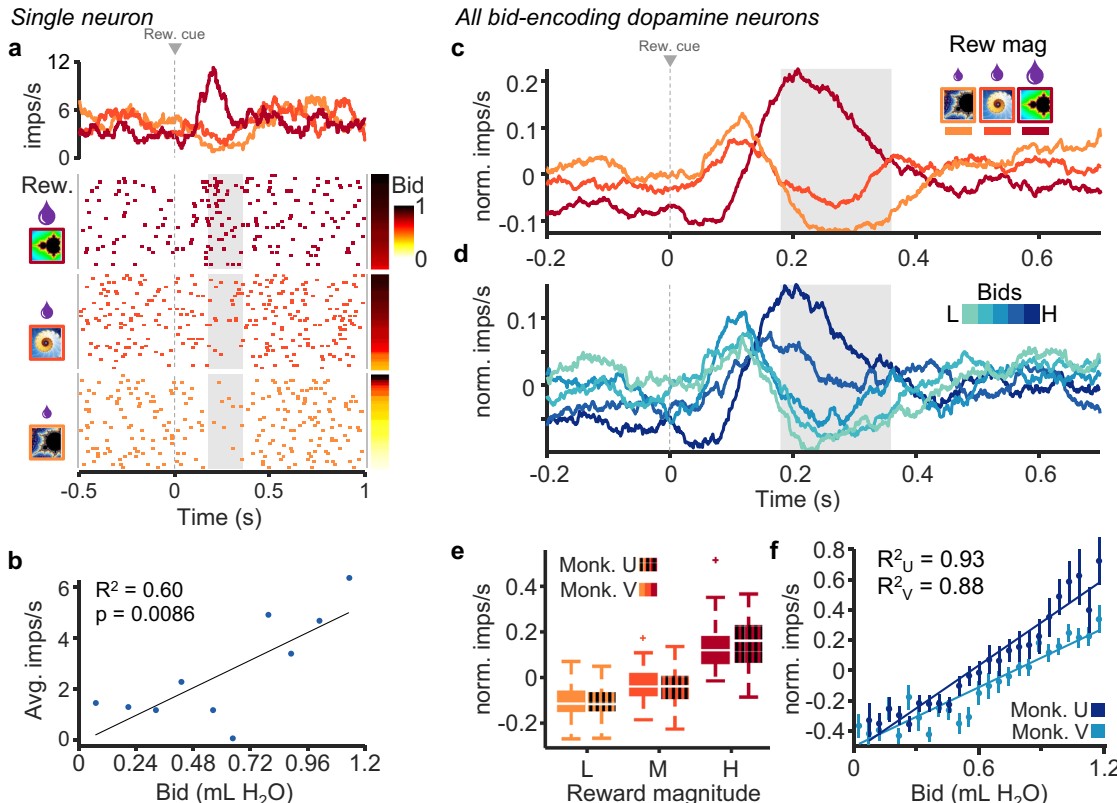

**Fig. 2 | Dopamine responses reflect subjective value on a trial-by-trial basis.**
**a** Peri-event raster and average impulse rate traces for a single dopamine neuron aligned to onset of fractal display. The raster is sorted by reward magnitude level (left) and monkey bid (right). **b** Correlation of average impulse rates per bin for monkeys' bids binned by tenths (simple linear regression). **c** Average traces of impulse rates of all bid-correlated dopamine neurons grouped by reward magnitude (Monkey V, $n = 41$ neurons). **d** Average traces of activity of all bid-correlated dopamine neurons grouped by bid quintiles (same neurons as in (**c**)). **e** Average normalized impulse rates during the post-event time period shown by the gray box in (**c**) (box center lines represent median, margins represent 25th and 75th

percentiles, whiskers represent the range of all non-outlier data, and "+" points indicate outliers; Kruskal–Wallis test, Tukey–Kramer multiple comparison test). **f** Mean normalized impulse rates shown in (**d**) with fitted regression lines (simple linear regression; error bars show standard error of the mean). For (**e** and **f**), $n = 32$ and $n = 41$ neurons for Monkeys U and V, respectively. Note that all analyses concerned only the second, value dopamine response component (gray analysis time windows in (**a**, **c**, **d**)), whereas the preceding first, attentional response component varied only inconsistently. Analysis time windows were 180 to 360 for monkey V, and 180 to 340 for monkey U (detailed in "Experimental Procedures"). These windows were used for these and all following analyses.

focused on the second, value component of dopamine neuron responses as described above). Dopamine responses showed good distinction between the three bid levels at each of the three reward magnitudes (Fig. 3c–e). While the responses with the high magnitudes reflected the positive reward prediction errors elicited by the high magnitude rewards (relative to the preceding predictive trial-start cross), the responses with the lowest magnitudes reflected mostly the elicited negative reward prediction errors, and the responses with the intermediate magnitude showed a mixture of positive and negative reward prediction error responses. Figure 3c–e shows that when the bids are evenly discretized into three levels (correcting for skewness of each distribution), there is good separation among bid levels. Thus, dopamine responses reflected the bids the animal placed within all three reward magnitudes independent of the externally represented reward cue. Single linear regressions (Eq. (4)) confirmed these results (Fig. 3f–h; Monkey V: $R^2_{high} = 0.32$, $p_{high} = 0.08$, $R^2_{mid} = 0.83$, $p_{mid} < 0.001$, $R^2_{low} = 0.27$, $p_{low} < 0.01$, $n = 41$ neurons; Monkey U: $R^2_{high} = 0.67$, $p_{high} < 0.001$, $R^2_{mid} = 0.74$, $p_{mid} < 0.001$, $R^2_{low} = 0.56$, $p_{low} < 0.01$, $n = 32$ neurons; for high, middle, and low bids, respectively). Traces for individual neurons are shown for each reward magnitude split by bid-tercile in Supplementary Fig. 6. Because higher reward magnitude elicited higher bids (Fig. 1d, e), we next asked whether the bid-sensitive neuronal responses might also reflect reward magnitude.

Same subjective value responses despite different reward magnitudes. Above we demonstrated that dopamine neurons encode subjective value (bids) prior to the bid being made. However, bid coding does not preclude reward magnitude coding in and of itself. To test whether reward magnitude was encoded independently of bidding, we examined the responses of neurons when bids were similar for two different reward magnitudes. Because there were low numbers of perfectly matched bids between reward levels, bids within 5% of one another were compared. Comparisons with significantly different bid distributions were eliminated (see "Methods" for complete explanation). For these similar bids (see "Methods"), we found no difference in dopamine responses between fractals indicating small vs. medium reward magnitudes (Fig. 4a), medium vs. large reward magnitudes (b), and small vs. large reward magnitudes (c) (analysis restricted to second, value response component, whereas the first, attentional dopamine response component varied inconsistently). Instead, the dopamine responses reflected the bids the animal made rather than the reward magnitude indicated by the fractals.

We next tested response differences for similar bids across *all* bid-encoding neurons as a group. For similar bids (<5%), we subtracted responses to lower reward magnitudes from responses to higher reward magnitudes; for this test, computed differences greater or lesser than zero indicate responses driven by reward magnitude. Concurrently, we found no difference in responses

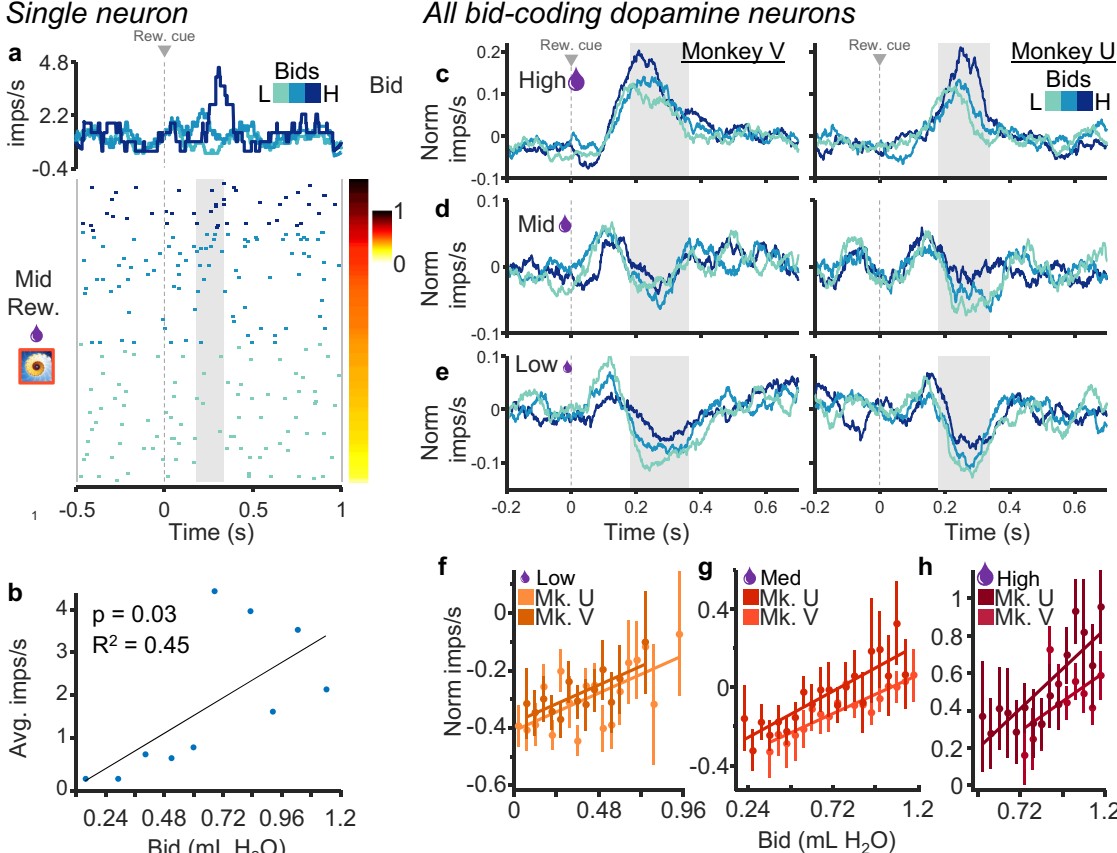

**Fig. 3 | Dopamine neurons exhibit graded responses that reflect the animal's bids irrespective of reward magnitude. a** Raster and peri-event average impulse rate for the middle reward magnitude only. The raster is sorted by bid (right) and the three shades of blue show the responses for thirds of the bid space (traces and raster). **b** Correlation with average impulse rates per bin for bids binned by tenths (simple linear regression). **c–e** Traces showing normalized impulse rates for low, middle, and high reward magnitudes individually; blue shades indicate thirds of the bid space as in (**a**). **f–h** Regressions of normalized impulse rates on bids for each

reward magnitude. Monkey V: $R^2_{high} = 0.32$, $p_{high} = 0.08$, $R^2_{mid} = 0.83$, $p_{mid} < 0.001$, $R^2_{low} = 0.41$, $p_{low} < 0.01$ ($n = 41$ neurons); Monkey U: $R^2_{high} = 0.67$, $p_{high} < 0.001$, $R^2_{mid} = 0.74$, $p_{mid} < 0.001$, $R^2_{low} = 0.56$, $p_{low} < 0.01$ ($n = 32$ neurons); for high, middle, and low bids, respectively. Error bars depict standard error of the mean). Note that all analyses concerned only the second, value dopamine response component (gray analysis time windows in (**c–e**)), whereas the preceding first, attentional response component varied only inconsistently.

between higher and lower reward levels for similar bids ($p > 0.05$ for each comparison; two-sided Wilcoxon signed rank test, $n = 41$ and $n = 32$ for Monkeys V and U, respectively) (Fig. 4d, e). Together, these data suggest that the dopamine neurons encoded the subjective value expressed by the bid rather than the reward magnitude indicated by the fractals.

**Dopamine responses decode future bids**

Given that the dopamine response to the fractal stimuli reflected the animal's bid rather than the reward magnitude indicated by the fractal, the question arose whether this neuronal response could decode the bid the animal was going to make a few seconds later. We addressed the question by using a Support Vector Regression (SVR) that decodes on a continuous scale, rather than binary distinctions typical for standard Support Vector Machine (SVM) classifiers. As the data derived from several weeks of recording, their non-simultaneous nature provided a rather conservative estimate of the decoding capacity of the dopamine response. We used the period of the second, value component of the dopamine response, as indicated by the gray areas in Figs. 2–4 (while neglecting the irrelevant attentional first dopamine response component). We trained the SVR on these neuronal responses from 80% of the bids, randomly selecting responses in each neuron from each tenth of the bid space. Using the remaining 20% of the bids and neuronal responses, we then tested the accuracy with

which the model predicted the monkeys' bids, using 300 iterations of 100 randomly selected trials (see "Methods").

When we added randomly selected responses from randomly selected bid-encoding dopamine neurons to the model, we found that decoding accuracy was low for single neurons but quickly increased to about 60% with ~20 neurons (Fig. 5, dark blue; $n = 41$ for Monkey V and $n = 32$ for Monkey U). The accuracy was lower when we included all other dopamine neurons (light cyan; $n = 145$ for Monkey V and $n = 123$ for Monkey U) and was lowest with only the non-bid-encoding neurons (light blue; $n = 91$ and $n = 104$ for Monkeys U and V, respectively), suggesting that the subjective reward value was largely encoded in the population of bid-encoding dopamine neurons.

While these data show the contribution of the "typical" dopamine neuron, we sought to assess the upper limit of decoding accuracy of dopamine neurons. For this aim, we added neurons to the model from best-encoding to worst-encoding, ordered by explained variance (R2 of individual bid-encoding dopamine neurons (Eq. (4)). We found that decoding accuracy reached asymptote with relatively few neurons in both animals (Supplementary Fig. 7a, b), suggesting high-fidelity encoding of subjective value in even smaller groups of dopamine neurons. The decoding accuracy did not improve by combining these neurons with non-bid-encoding neurons (light cyan in Supplementary Fig. 7a, b). As anticipated, when we added bid-encoding dopamine neurons in the reverse order, from the worst-encoding to the best-

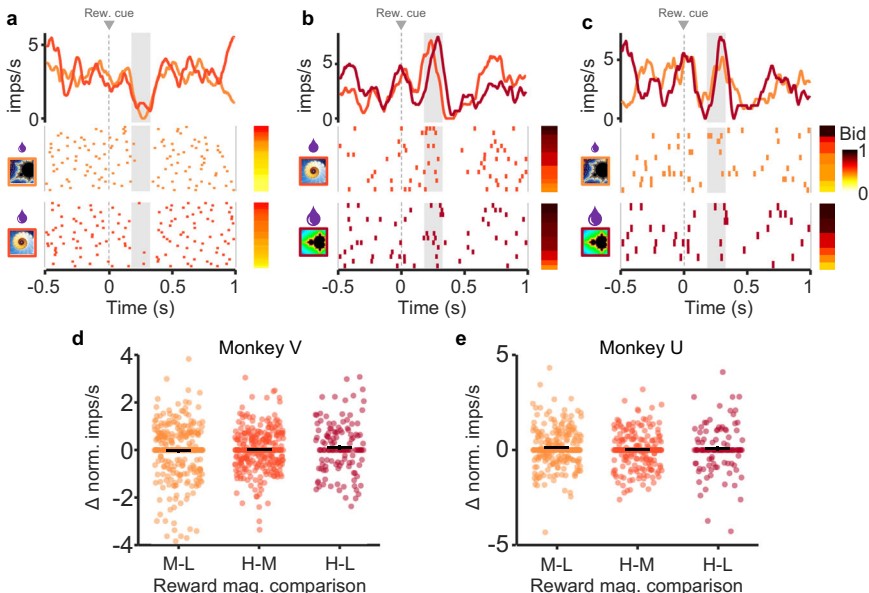

**Fig. 4 | Dopamine responses exhibit similar responses with similar bids despite different reward magnitudes. a–c** Raster plots and peri-event averages for similar bids for low, middle, and high reward magnitudes, respectively. Rasters are sorted by bid (bar to the right of rasters); different shades represent lower vs. higher reward magnitudes for each comparison (left of rasters). **d, e** Difference measure of normalized responses for each comparison for Monkey V and Monkey U. No

significant differences with any comparison ($p > 0.05$, two sided Wilcoxon signed rank test; $n = 41$ and $n = 32$ for Monkeys V and U, respectively). Note that all analyses concerned only the second, value dopamine response component (gray analysis windows in (**a**–**c**)), whereas the preceding first, attentional response component varied only inconsistently.

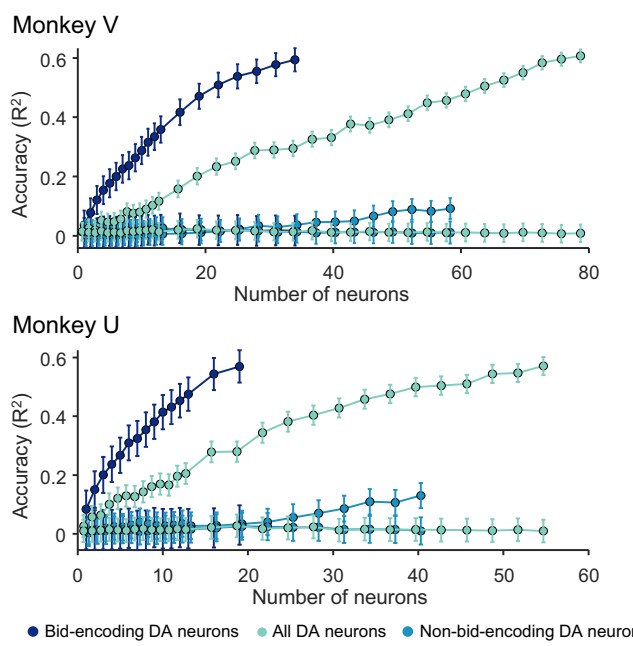

**Fig. 5 | Results from analysis by support vector regression (SVR) for the two monkeys.** The model was trained on neuronal responses and bids using 80% of the data. Bids were predicted from responses with the remaining 20% of the data (five-fold cross-validation method). Bid prediction accuracy is shown as R2 (ordinate; each data point shows mean ± standard error of the mean from 300 iterations) for bid-encoding dopamine neurons (dark blue; DA; $n = 32$ for Monkey U and $n = 41$ for Monkey V), all dopamine neurons (second curve from top, light cyan; $n = 123$ and $n = 145$ for Monkeys U and V, respectively), non-bid-encoding dopamine neurons (light blue), and shuffled data (three flat curves at bottom in corresponding colors; $n = 91$ and $n = 104$ for Monkeys U and V, respectively). Neurons were added to the model in random order (see "Methods").

encoding, decoding accuracy fell below that of the average dopamine neurons (Supplementary Fig. 7c, d), suggesting that most of the decoding accuracy was derived from the best bid-encoding neurons and that little was gained from the worst bid-encoding neurons (the last neurons in Supplementary Fig. 7c and d were the first neurons in **a** and **b**, respectively).

In conclusion, the SVR demonstrated limited fidelity of coding in single neurons that improved rapidly with small populations. The high accuracy is remarkable given that decoding continuous behavior, such as bidding in the BDM, is more challenging than traditional classification of binary choices.

## Discussion

These data show that the phasic dopamine reward signal encodes subjective reward value on an instantaneous, trial-by-trial basis. We used the BDM auction-like mechanism to estimate subjective reward value in an accurate manner in each trial (incentive compatibility) irrespective of the value being dictated by externally presented reward information (Fig. 1). We found that the reward responses of dopamine neurons followed the animal's fluctuating BDM bids (Fig. 2), both with constant and with varying reward amounts (Figs. 3 and 4). The systematic and consistent bidding (Fig. 1c–e) and the correspondingly monotonic increase of dopamine responses with reward amounts and bids (Fig. 2c, d) argue in support of subjective reward value coding rather than objective amount coding (Fig. 3c–h) and, importantly, argue against other, non-value factors, like distraction of the animal or random bidding. The SVR decoder predicted the BDM bids accurately from the dopamine responses, which demonstrates the validity of the neuronal code for subjective reward value (Fig. 5). Thus, the dopamine responses encoded the instantaneous subjective reward value on a trial-by-trial basis, rather than the average subjective or objective (physical) reward value.

The BDM is often considered a key estimation mechanism for revealing the subjective reward value by encouraging subjects to

truthfully report their valuation, a property called incentive compatibility. Exaggerated bids would allow the price to rise beyond the bidder's own private value, whereas understated bids incur the risk of losing out on the desired good. In this way, BDM bids reveal the instantaneous fluctuations of true subjective reward value. Nevertheless, it should be kept in mind that BDM bidding in humans is often fraught with incomplete understanding of the mechanism, context sensitivity, framing, and incomplete training[20]. We addressed such concerns in the current experiment by extensively training and testing our two monkeys in tens of thousands of trials in several BDM implementations before starting the neuronal recordings. As a result, both animals showed consistent performance during the several months of neuronal recordings, demonstrating well-ordered bids in thousands of trials (Fig. 1c–e), as also shown previously[20]), which limits these potential issues.

Formal economic utility represents subjective reward value as a mathematical function of reward amount[21–23]. A previous study has shown that dopamine neurons code subjective reward value as mathematically defined utility[6]. Similarly, mathematically defined temporal discounting functions represent the subjective value of delayed rewards, and dopamine neurons code such subjective value of delayed rewards[4]. However, subjective value extends beyond mathematically defined utility and temporal discounting functions and refers in a more general sense to the satisfaction and benefit obtained by rewards. Such subjective value is captured in a rather direct and instantaneous way by BDM bids. Thus, the current experiments using BDM confirm the subjective reward value coding by dopamine neurons and extend it to trial-by-trial measures.

The observed variation of BDM bids within and between sessions for the same reward magnitude demonstrated the subjective nature of the reward valuation (Supplementary Fig. 2a). While being subjective, the animals' bidding was meaningful, as evidenced both by the overall larger bids for higher reward magnitudes and the coherence of bids among magnitude levels (Fig. 1c–e, Supplementary Fig. 2b, c). The small effect of previous bidding outcome (win/loss; Fig. 1e) and win/lose streak may reflect an instantaneous or short-term adaptation. This effect, and other challenges to accurate BDM bidding, such as possible task misperceptions[24], may warrant further analyses in future behavioral primate studies on BDM. The subjective nature of BDM valuation was also apparent in inter-individual valuation differences, as satiety and win/lose streak varied bidding substantially between the two monkeys (Supplementary Fig. 2e, f). Together, these results confirmed that the animals' bids in our BDM reflected the subjective value for the tested juice volumes generated by the brain from external reward information.

The reported dopamine reward prediction error responses to the fractal stimuli seemed to follow both the BDM bids and the externally indicated reward amounts (Fig. 2). However, the dopamine responses varied with the bids even when reward amounts were held constant (Fig. 3), and the responses were similar with similar bids for different reward amounts (Fig. 4). Thus, the dopamine responses reflected the bids and not the reward amounts, suggesting coding of subjective reward value distinct from reward magnitude.

The SVR results demonstrate that dopamine responses to the reward cues reliably predicted the subsequent BDM bids made by the animal (Fig. 5). By adding randomly selected dopamine responses to the decoding model, the low decoding accuracy increased quickly to about 60% with 20–30 neurons. Adding neurons from best-coding to worst-coding confirmed the small neuron numbers required to accurately decode the bids (10–20 neurons) (Supplementary Fig. 7a, b). As previous studies have shown that dopamine excitation drives behavior in monkeys and rodents[25–29], our data would suggest that the dopamine response to the fractals may be instrumental for generating useful bids; only a small population of neurons (10–20) would be necessary for this effect. However, while the shuffled data

resulted in an accuracy of 0%, the accuracy of 60% with the SVR is much smaller than the 80–90% accuracy achieved with binary decoders such as nearest neighbor, linear SVM, and discriminant analysis classifiers[30–34]. A possible explanation for the lower accuracy of SVR as compared to a binary classifier may lie in the continuous nature of the SVR, and that accuracy is possibly further reduced due to binning the bids necessitated by our limited number of trials per bid. Thus, the observed decoding accuracy may represent a conservative estimate of the predictive capacity of the neuronal responses. While these quantitative considerations are technically important, the observed decoding of bids from dopamine responses confirms the validity of the dopamine signal for coding subjective reward value expressed behaviorally by the bids, which is important as the bids are made without a constraining external option set.

A critical question that remains unexamined is whether dopamine neurons encode the value of affective or socially relevant stimuli in a similar manner. Although such stimuli have subjective value[35–37], it is unclear whether value is ascribed in the same way as with stimuli representing physical nutrient rewards. Further work is needed to better understand how nutrient rewards are distinguished from affective/social rewards and whether dopamine neurons encode these stimuli in similar ways.

Taken together, these data demonstrate the ability of dopamine neurons to encode subjective value in an instantaneous, trial-by-trial manner. The auction-like BDM mechanism provided an appropriate behavioral mechanism for rapidly revealing the animal's subjective valuation. The SVR decoder demonstrated that the dopamine signal was capable of predicting the next BDM bids, thus adding to the suggestion of a valid and precise neuronal code for instantaneous subjective value.

## Methods
### Animal ethics, welfare, and surgical implantation
We used two adult male rhesus monkeys (*Macaca mulatta*; Monkey V: 11 kg and Monkey U: 17.5 kg) that were born and raised at the Centre for Macaques at Porton Down, UK. This research has been ethically reviewed, approved, regulated and supervised by the following UK and University of Cambridge (UCam) institutions and individuals: UK Home Office, implementing the Animals (Scientific Procedures) Act 1986, Regulations 2012, and represented by the local UK Home Office Inspector, UK Animals in Science Committee, UK National Centre for Replacement, Refinement and Reduction of Animal Experiments (NC3Rs), UCam Animal Welfare and Ethical Review Body (AWERB), UCam Biomedical Service (UBS) Certificate Holder, UCam Welfare Officer, UCam Governance and Strategy Committee, UCam Named Veterinary Surgeon (NVS), and UCam Named Animal Care and Welfare Officer (NACWO).

The two monkeys were housed in adjoining cages and placed on a restricted water regimen calibrated by body weight. Behavioral data were acquired from both animals for a prelusive publication[16]. Each weekday, we transported the animals to the experimental laboratory in an individually adjusted primate chair (Crist Instruments). Animals sat in this chair for the duration of the daily tests, which never exceeded 5 h. We provided animals with fruit and vegetable enrichment on Friday evening and ad-libitum access to water throughout Friday evening and Saturday.

We implanted Monkey U with a titanium headpost (Crist instruments) used for head fixation and later implanted a recording chamber after the headpost had integrated with the bone. For Monkey V, we implanted head-fixation hardware concomitantly with the recording chamber. Chambers were centered on the skull laterally using a stereotaxic head holder and a Kopf stereotaxic manipulator. After recovery from surgery, we drilled craniotomies above recording sites, and chambers were monitored and cleaned daily. Once experiments were completed, recording sites were marked with electrolytic lesions

(15–20 μA, 20–60 s). Upon completion of the experiment, we sacrificed the animals by administering an overdose of sodium pentobarbital (90 mg/kg, IV) and subsequently perfused with 4% formalin in 0.1 M phosphate-buffered saline. We confirmed recording positions histologically from 40 μm slices stained with cresyl violet.

## Experimental setup

During experiments, animals were head-fixed while seated in a primate chair. All experiments were performed in a dimly lit experimental isolation booth (Crist instruments) to minimize disruption. The monkeys were positioned so that their eyes were ~70 cm from a computer monitor. The joystick lever was attached to the chair and made accessible via ~15 cm2 opening in the front of the chair. Water and juice were delivered through separate spouts positioned ~5 mm from the animal's mouth. Fluid delivery was controlled by gravity-fed solenoid valves connected to 1 L beakers using silicone tubing. Juice and water delivery valves were calibrated to deliver precise volumes (SD < 0.01 ml). The monkeys were trained to use a custom-made touch-sensitive joystick (Biotronix Workshop, University of Cambridge) to interact with the task displayed on a computer monitor as previously described[16]. The joystick lever was mounted on the right-hand side of the animal and was only movable forward and backward, resulting in upward and downward movement of the cursor that indicated the monkey's bid (upward was higher bid). The rightward position of the joystick lever corresponded to the rightward position of the bid cursor on the computer monitor. The constant rightward position of lever movement and bid cursor precluded any side bias in bidding and neuronal responses.

## BDM elicitation of subjective value

The BDM is a second-price sealed-bid auction-like mechanism that has been shown to elicit truthful estimates of subjective value on a single-trial basis. Typically, a BDM bidder will garner the highest payoff by bidding exactly the value they place on the good. Economists refer to this as incentive compatibility: the optimal strategy is to bid one's true subjective value. Bidding too high (overbidding) increases the risk of overpaying, and bidding too low (underbidding) increases the risk of not obtaining the desired item (Fig. 1a, b)[38]. Three features are key to the BDM's incentive compatibility: (1) the second-price nature is essential for revealing the true subjective value because the opponent's bid is unknown; it prevents overbidding because the unknown bid of the opponent may exceed the value and thus result in overpaying; it prevents underbidding because the opponent might outbid them, and (2) the bids are hidden until all bids have been submitted (sealed bid auction). Thus, BDM is akin to a private value auction, as subjects are not able to infer the value other bidders place on the good. As opponent bids are drawn randomly from a uniform distribution, a common value cannot be surmised, even with consecutive trials. Many details of the extensive experience of several years in BDM bidding of the animals used in this study have been presented before[16], and only the behavioral results relevant for the current neuronal analyses will be described here.

The described characteristics explain our rationale for using the BDM. Any study of subjectively determined reward value requires that the measured events, namely the bids given by the animal, reflect the true subjective value at each moment. The incentive-compatible nature of the BDM provides exactly that assurance. Further, the BDM does not require a biological opponent, which makes the experimentation less complicated and simplifies the interpretation of neuronal data by avoiding confounds from an opponent's behavior.

Recent experiments demonstrated that rhesus monkeys can show meaningful performance in behavioral tasks implementing a BDM[16]. On every trial, the monkey bid against a computer opponent using a joystick; the animal paid from a water endowment that had been allocated on every trial with the same amount. If the animal's bid equaled or exceeded the computer bid, the animal won the auction and paid the price defined by the computer bid (second-price nature of BDM); thus, the animal received the juice it had bid for, plus the rest of the water endowment after subtraction of the computer bid. If the animal's bid was below the computer bid, it lost the auction and received the full water endowment (Fig. 1b). To increase the number of trials per reward magnitude, monkeys bid for only 3 reward magnitudes. The range of reward magnitudes was calibrated to each animal's preferences so that the full range of bids was well represented. In addition, monkeys bid for fixed goods rather than for lotteries, which avoided confounds from the animal's risk attitude. Our task also circumvented the endowment effect (a tendency to over-value previously acquired goods), as monkeys do not "pay" an amount already acquired but rather indicate the amount of water they are willing to forgo from water paid out at the end of the trial. Specifically, monkeys receive a water payout on every non-error trial (see below) regardless of whether they win or lose the auction; their bids reflect how much water they are willing to forego on each trial to get the juice reward.

## BDM Task

BDM trials were initiated with a yellow cross at the center of the screen (Fig. 1a). After 0.5 s, a fractal representing one of three different juice volumes appeared. After 1.0 s, a vertically oriented rectangle appeared that denoted the bid space that was defined by the smallest and largest reward amount the animal could bid for (0 and 1.2 ml; hashed black lines on white background). A bid cursor overlayed the bid space rectangle (magenta). Forward and backward movement of the joystick generated up and down movements of the cursor within the bid space. Bidding had to be initiated within 0.5 s and stabilized within 5 s, otherwise, the trial was terminated and the screen briefly flashed red indicating a failed trial and a wait penalty equal to the remaining trial time plus 2 s. Letting go of the joystick at any point during the trial or moving during any period except the bidding epoch also resulted in trial termination. The total bid space represented 1.2 ml of water; monkeys' bids indicated how much water they were willing to sacrifice for a given fractal. Once the monkey's bid was stable for more than 0.5 s, the bid of the computer opponent appeared and the direction of the hashed lines below the computer bid reversed, indicating the amount of water to be "paid" for obtaining the juice if the animal won the BDM (second price). If the monkey won the BDM, the juice was paid out and the fractal would disappear from the screen after 1 s, followed by water payout that lasted up to1 s (1.2 ml minus the computer bid measured in ml, which reflected the second price character of BDM) and removal of the bid space from the screen. If the monkey lost the BDM, the fractal disappeared and the water was paid out in full (1.2 ml).

While the three fractals shown in Fig. 1a were used for the current neuronal recordings, the same two monkeys had been trained with five different fractals for five juice volumes[16]. The monkeys showed consistent bidding across these different stimulus sets, which suggests that the use of the three specific fractals in the present experiment should not have affected the neuronal results.

## Behavioral analysis

BDM bids reflect subjective value and change from trial to trial. This can be demonstrated formally for expected-utility maximizers[38] and has been supported empirically in rhesus monkeys in previous work from our laboratory[16]. Here we sought to further test whether changes in bidding resulted from changes in value or from other value-irrelevant task features. To test which elements of the task contributed most to bid variance, we first used a cross-validated lasso regression (lasso function, Matlab, Mathworks) to identify variables that contributed to bid variability. Lambda (tuning parameter) was selected by taking the value corresponding to one standard error above the mean squared error based on 2000-fold cross-validation (Supplementary Fig. 2d). We used the following 29 regressors in the lasso model: 1)

reward value, 2) starting bid, 3) previous total liquid, 4) day of week, 5) session number, 6) previous trial failure, 7) previous trial result, 8) previous result from trial with same reward magnitude, 9) trial number, 10) competing bid t-1, 11) competing bid t-2, 12) competing bid t-3, 13) competing bid t-5, 14) competing bid t-7, 15) competing bid for the same reward magnitude t-1, 16) competing bid same reward magnitude t-2, 17) competing bid same reward magnitude t-3, 18) competing bid same reward magnitude t-4, 19) competing bid same reward magnitude t-5, 20) competing bid same reward magnitude t-6, 21) competing bid same reward magnitude t-7, 22) competing bid same reward magnitude t-8, 23) competing bid same reward magnitude t-9, 24) competing bid same reward magnitude t-10, 25) average of competing bid same reward magnitude t-2 to t-1, 26)) average of competing bid same reward magnitude t-3 to t-1, 27) average of competing bid same reward magnitude t-4 to t-1, 28) average of competing bid same reward magnitude t-5 to t-1, 29) average of competing bid same reward magnitude t-6 to t-1, 30) win streak for same reward value, 31) lose streak for same reward value.

Variables remaining in the correspondent model were then used in a mixed-effects model to identify which had the largest impact on bidding behavior (Fig. 1e).

$$y = \beta_{0ij} + \beta_1 RewardMagnitude_{ij} + \beta_2 StartingBid_{ij} + \beta_3 TotalLiquid_{ij}$$
$$+ \beta_4 PreviousCompetingBid_{ij} + \beta_5 PreviousResult_{ij} + b_{0ij}$$
$$+ b_{i1} TrialNumber_{ij} + b_{i2} SessionNumber_{ij} + \varepsilon_{ij}$$

$$(1)$$

Because reward magnitude is highly intercorrelated with the subjective value and therefore with the bid, we sought to determine what other factors, besides reward, most prominently predicted changes in bids. For this, we used a reduced mixed-effects model with reward magnitude included as a random effect (Supplementary Fig. 2e).

$$y = \beta_{0ij} + \beta_1 StartingBid_{ij} + \beta_2 TotalLiquid_{ij} + \beta_3 PreviousCompetingBid_{ij}$$
$$+ \beta_4 PreviousResult_{ij} + b_{0ij} + b_{i1} RewardMagnitude_{ij}$$
$$+ b_{i2} TrialNumber_{ij} + b_{i3} SessionNumber_{ij} + \varepsilon_{ij}$$

$$(2)$$

We also sought to determine the influence of consecutive wins or losses on the current bid independent of reward magnitude. Because consecutive wins/losses were intercorrelated with the previous result (win/lose for t-1), we performed a separate analysis including the previous result as a random effect.

$$y = \beta_{0ij} + \beta_1 StartingBid_{ij} + \beta_2 TotalLiquid_{ij} + \beta_3 PreviousCompetingBid_{ij}$$
$$+ \beta_4 PreviousWinStreak_{ij} + \beta_5 PreviousLoseStreak_{ij}$$
$$+ b_{0ij} + b_{i1} RewardMagnitude_{ij} + b_{i2} TrialNumber_{ij}$$
$$+ b_{i3} SessionNumber_{ij} + b_{i4} PreviousResult_{ij} + \varepsilon_{ij}$$

$$(3)$$

If changes of individual bids reflect changes in subjective value across trial, then these changes should be apparent in bids across all three reward magnitudes. We tested this by measuring bid coherence within and between sessions. For within-session coherence, bids were interpolated for each reward magnitude to create three equally populated vectors of bids to retain trial-by-trial temporal fidelity. Each vector was then correlated with the others in three separate tests comparing low reward magnitudes to mid magnitudes (L:M), mid magnitudes to high magnitudes (M:H), and low magnitudes to high magnitudes (Supplementary Fig. 2b). The resulting rho values provide a relative estimate of coherence with values above zero indicating

positive coherence. Note that because values were interpolated for trials where a given reward magnitude was not represented, this analysis can only provide a lower bound for the estimated coherence.

### Electrophysiological recording and analysis

Electrophysiological signals were recorded using electrodes made in house or ordered from Alpha-omega (125 μm diameter, 60-degree bevel). Electrodes were loaded into a sterile 23-gauge stainless steel cannula which was used to pierce the meninges and stabilized the electrode's path through the brain. Electrodes were lowered into the midbrain using an electrode micromanipulator from Nan instruments (model: CMS) or Narishige (model: MO-97). Recordings were amplified and band-pass filtered from 100 to 5000 Hz (custom hardware and Bak Electronics). Recordings were digitized using a National Instruments data acquisition card and visualized with custom Matlab (Mathworks) software. Neuronal impulses were sorted offline using Spike2 version 7.8 (Cambridge Electronic Design).

### Ventral midbrain localization

Coordinates for the recording sites in the ventral midbrain were determined using sagittal radiograph images of the head in a stereotaxic frame and electrophysiological signatures of surrounding cell groups. Specifically, animals were placed in a stereotaxic frame and a cannula was inserted in the center-most position of x-y plane of the recording chamber. Bone features (interaural origin and the clinoid process of the sphenoid bone) were used to determine the approximate anteroposterior and dorsoventral positions of familiar nuclei. Using these positions as anchors, stereotypical electrophysiological responses from the red nucleus and ventral posteromedial nucleus of the thalamus guided the localization of the recording sites of dopamine neurons in the ventral midbrain.

### Statistical analysis of dopamine neuron responses

Dopamine neurons were identified using canonical criteria: a wide impulse waveform (>1.8 ms), low baseline impulse activity (<10 Hz), and consistent responses to unpredicted reward delivery. All neuronal impulse data were binned in 1 ms bins for analyses. Population analyses were performed with all dopamine neurons; analyses of bid-encoding neurons were performed using only dopamine neurons that exhibited a positive correlation with the monkeys' bids (see below). All statistical analyses were performed on raw activity (for single neurons) or z-normalized activity from time windows defined by gray boxes in figures (for all bid-encoding neurons and population analyses). The time ranges of the gray boxes were estimated differentially for each animal to account for differences in response latency and duration. The gray boxes start at the end of the first component of the dopamine response and end when the signal returns to baseline. Specifically, because dopamine responses typically exhibit two components[19], analysis time windows were defined for each animal so as not to include the initial "attentional" component of the response and to only regard the second "value" component. In order to establish when the first component was finished, we examined the lowest value reward as we expected that it would produce a negative prediction error during the second "value" component, but would not affect the positive-going signal produced by the first "attentional" component. The start of the analysis window was defined when the initial component of the averaged signal returned to the mean of the baseline (0–500 ms before the fixation cross), rounded to the nearest 20th ms. To establish the end of the analysis window, we used the largest reward value as we predicted that it would produce a well-defined positive prediction error signal. The end of the analysis window was defined when the average signal was no longer significantly different from zero (20 ms bins were tested between $t = 0$ and $t = 500$ ms; Wilcoxon Sign-rank, $p < 0.05$, Bonferroni-Holm correction). These windows were 180–360 ms for monkey V and 180 to 340 ms for monkey U. Average traces shown in

figures were smoothed with an 80 ms or 100 ms moving average. For group-level analyses, data were z-scored to account for variance among neuronal activity. Bids were discretized into bins for individual neuron analyses and for group-level analyses to obtain more accurate estimates of average responses for a given bid-range (e.g., by splitting the overall bid range into tenths and averaging the neural responses within bins; for specifics, see Results).

Correlations between bids and neuronal responses were assessed using the following linear regression:

$$y = \beta_0 + \beta_1 Monkey\ Bids \qquad (4)$$

To test whether subjective value, as expressed by the bids, was driving changes in activity independent of reward magnitude, responses were correlated with bids when reward magnitude was held constant (Fig. 3a–h). This was further tested by comparing neuronal responses for matched bids between reward magnitude levels for individual sessions (Fig. 4). For this analysis, for each experimental session, responses corresponding to similar bids (within 5%) made for two different reward magnitudes (medium vs. low, high vs. medium, high vs. low) were pooled across all neurons and compared using a paired Wilcoxon sign-rank test. Significant differences in neuronal activity in this test would be indicative of responses driven by reward magnitude independent of bid. No difference indicates that neuronal responses were driven by bids independent of reward magnitude.

### Neural decoding with support vector regression (SVR)

We used Linear SVR to predict the monkeys' bids based on the responses of dopamine neurons to the fractals. The continuous bids were discretized into 10 non-overlapping bid ranges from [0–0.1] to [0.91–1.0]. The model was subsequently trained on neuronal responses from each bid range. For each neuron, 10 trials were selected at random, providing 100 randomly selected trials (Supplementary Fig. 7). The SVR model was adapted from earlier used methods[39] and implemented with custom-written code in Matlab 2021b using "fitrsvm" and "predict" functions for model training and testing.

We used three separate SVR models to assess how well dopamine neurons could predict animals' bids. Neurons were added to these models randomly or by order of explained variance for monkeys' bids. Neurons were added to model 1 from highest to lowest correlation, in model 2 from lowest to highest correlation, and randomly in model 3. Models one and two provide a lower and upper limit of decoding accuracy, and model 3 allows for comparison with similar analyses in previous works. Model performance was assessed with coefficient of determination $R^2$ (explained variance). Each model was tested against shuffled bid and neural response data. The binary differences between $R^2$ coefficients obtained for real data against the shuffled data were verified by Wilcoxon rank-sum test ($p < 0.01$).

Each SVR model was trained/tested using a five-fold cross-validation (80% / 20%) method: eight trials from every bid category (8 trials × 10 categories) were used for training, and the remaining two trials from every bid category were used for testing (2 trials × 10 categories; Supplementary Fig. 8). This procedure was repeated five times, thus providing five $R^2$ estimates for each set of 100 randomly selected trials. The reported explained variance $R^2$ for model-predicted versus actual bid data was calculated by averaging the $R^2$ values from 300 iterations of the 100-trial random selection procedure described above.

### Reporting summary

Further information on research design is available in the Nature Portfolio Reporting Summary linked to this article.

## Data availability

All processed data generated in this study have been deposited in the Figshare database and can be accessed using the following https://doi.org/10.6084/m9.figshare.25734288.

## Code availability

All code used for analyses in this study has been deposited in the Figshare database and can be accessed using the following https://doi.org/10.6084/m9.figshare.25734288.

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

## Acknowledgements
We thank Aled David and Christina Thompson for animal and technical support and Alexandre Pastor-Bernier, Raymundo Baez-Mendoza, Fabian Grabenhorst, William R. Stauffer, John O. Ledyard, Charles R. Plott, and David M. Grether for discussions on the design of this experiment. This study was supported by Wellcome Trust (WT 095495, WT 204811, WT 206207; WS), European Research Council (ERC; 293549; WS), and US National Institutes of Mental Health (NIMH) Caltech Conte Center (P50MH094258; WS). For the purpose of Open Access, the authors have applied a CC BY public copyright licence to any Author Accepted Manuscript version arising from this submission.

## Author contributions
D.F.H., A.A.M., and W.S. designed the study, D.F.H. and R.W.H. performed neurophysiology experiments, A.A.M. performed foundational behavioral experiments, D.F.H. analyzed data and constructed figures, A.S. performed SVR analysis, D.F.H., and W.S. wrote the paper.

## Competing interests
The authors declare no competing interests.
