## [Peer Review File · Nature Communications]

Dopamine neurons encode trial-by-trial subjective reward value in an auction-like taskREVIEWER COMMENTS

Reviewer #1 (Remarks to the Author):

In this manuscript, the authors make use of a well-defined and heavily validated behavioral task couched in economic theory – the Becker-DeGroot-Marschak second-price auction (BDM) - to identify the neural correlates (dopamine (DA) neuronal firing) of subjective value estimates. Previous definitions of value and/or preference have often relied on modeling decisions across multiple trials to estimate value. Here, the authors put forth a method that requires no modeling or estimation; the methodology and data demonstrate that immediate value can be quantified in the firing patterns of DA neurons in the minds of those making value estimates.

The findings are highly significant: DA neurons encode, not the discrete external-world values of cues assigned by investigators, but the internal subjective values that individuals themselves experience on a moment-by-moment basis.

The work quantitatively associates the notion of value, often assigned by investigators designing and carrying out neuroeconomic paradigms, with subjects' own internal estimates of value, and therefore brings the field closer to understanding the neural correlates of preference. The authors may consider mentioning preference explicitly in order to broaden the interpretation of their findings.

My one significant comment about the data and the figures; There appears to be activity prior to the grey shaded region in the figures. It is very interesting but I do not feel it is adequately described or addressed. There are some comments below that pertain to this and the use and definition of the gray shaded region in the figs.

As for more potential analysis of the currently available data: the authors mention a correlate with win/lose bidding. Does winning or losing a bid increase or decrease subsequent bidding? I think finding a way to look at bidding across trials would be very interesting. Note: I do see that there is an effect of previous bid and result (Fig 1e), but how might one interpret what is going on across multiple trials? Do multiple losses occur in a row, multiple wins; does this influence bidding?)

I think the discussion section could be improved; there are important points therein but it feels a little disorganized.

The authors might consider future studies documenting DA responses to naturalistic affective stimuli, including images of predators or social stimuli. Typically these stimuli are valued, but are often not paired with reward in a learning context.

The following is a list of edits and in some cases stylistic suggestions.

Figures

A general comment related to task diagram and subsequent plots of neural activity. Would you please explicitly put task timing info on the figure in 1a? I don't see that in the text or figure legend. This is related to second point: I think all your plots are aligned to fractal on time, but I only found that in 1 obvious place, the legend of fig 2. You might explicitly put on each t=0 line the text "fractal on" or "cue offered" or something of the like.

What is the gray shaded region in the neural plots? It says "post-event time period" but what is that and how was it defined? I assume this is your ROI, but how was it selected? It looks like there is information encoded in the period before that grey shaded region in Figs 2c, d and 3c-e. When is the fractal cue turned off / or bid line offered to the monkey?

The t=0 line is very faint in figs 2a; missing in 2c and d; faint in 3a; missing in 3 c-3; missing in Fig 4a-c. You may or may not want to include it in all figs.

In summary, please consider how best to display task timing info in Fig 1a and how to highlight the relevant epochs in the remaining figs (and clarify what the gray region is and how/why it was selected).

Supplemental Figures

For the indicated prediction error and task events shown in S1; does the indicated event refer to the start and subsequent epoch? Is the reward cue a prediction error event (and not task event)? Perhaps clarify these definitions? And consider using visual braces to indicate the epochs of interest.

Line 67

Why not refer to preference here? Stimuli and outcomes we "like" - independent of secondary reinforcers - seem more akin to the internal values you are investigating than the mentioned references about musical tunes and imagined speech. Or say: "Neuronal signals for internal processes exist. For example..." Is there not a literature on neural signals related to viewing and rating fine art, or attractive faces etc?

Line 82

Missing a word here, probably "us" or "one"

Line 91

The phrasing "externally imposed" seems somewhat awkward. Alternative: "... than values defined by external task constraints"?

Line 112

I don't believe there is text in the ms to describe figure 1b (but do see it in the figure legends. Please briefly describe Fig 1b in the ms text.

Line 119

Are you saying that the SD of bidding should be not statistically different for the 3 volumes – if driven by internal value? I don't follow. How is the variance an indicator of internal bidding (fidelity)? There are likely motivational effects for the three different reward volumes, No? I do see your analysis of coherence (supplemental 2b), but the distributions on the right in the small panel of Fig 1c look different. Do these curves look different because of bidding over time? It looks like it comes up with the high and mid magnitude cues.

I believe your demonstration that bid variability results from changes in internally generated subjective values, but how this relates to coherence, or is verified by bid coherence, is unclear to me.

Line 131

You say: "factors that significantly affected the animal's bids...". But what about the bids? Their quantitative value?

Line 132

The win/lose effect is intriguing! What is the direction of the effect? Is this shown or addressed somewhere (I have overlooked)? Does winning or losing a bid increase or decrease subsequent bidding? Either for the same cue or any cue?

Line 152

“... cue for water payout...” how do you know it is this cue, in particular? From the task diagram in Fig 1a, it looks like there is a lot of available info at that point in the trial.

Line 160

does this neuron show suppression for medium and low reward magnitudes – during the grey shaded region? I am confused about the gray shading. It looks like DA-neuron activity encoding for small and medium peaks before the large reward cue peaks? This is interesting; do you have thoughts about it?

Suppression during gray shaded time period?

Why the early peak for mid and low volumes and later peak for max volume?

Fig 2d

note early peak for low bids and later peak for high bids - why? Have you done some sort of temporal analysis to identify when DA neurons encode the internal bid estimates?

Line 191; Fig 3d, 3e

Still - lower bids show earlier spiking! Especially in mk V. What does this mean?

Peter M Kaskan

Reviewer #2 (Remarks to the Author):

This paper used an economic method, the BDM auction, to assess subjective trial by trial values of options, to monkeys. The authors then assess values of offers in a computerized task and ask whether dopamine responses correspond more closely to subjective or to objective value. They find the answer is subjective value.

These results are not particularly surprising or important. There have been dozens of studies asking whether neurons or hemodynamic responses more faithfully encode subjective or objective values, going back over 20 years. In every single case, the answer was always exactly the same: the brain encodes subjective values, not the objective values. The authors here find exactly the same thing. So there's nothing surprising about these results, nor do they change existing theories, nor do they provide needed support for any proposed, or even generally accepted but unsubstantiated theories.

The interesting thing about the paper is the use of the BDM task, but, of course, that is not novel here. The authors have already written that method/result up in a J Neuroscience paper last year.

There's another, deeper problem with that the paper that the authors seems to be unaware of, so I will mention it here. The BDM only works given certain assumptions that are almost certain to be violated in this case (see, e.g., Plott and Zeiler, 2005). These include (1) the idea that the subjects fully understand the structure of the BDM auction (Chou et al., 2009), (2) that they fully trust the experimenter and the stability of the task, and (3) that they are risk-neutral, (4), that subjects have access to their own internal values and (5) that variability in performance is due to variability in preference not things like task engagement, distraction, etc.

Indeed, the BDM, while great in theory, ore often than not fails in practice with humans - and monkeys are sure to be worse (for a good recent review of this topic, see Bull, Courty, Doyon, and Rondeau, 2019).

For (1), I encourage the authors to read Cason and Plott (2014), which deals with this issue, and demonstrates notable failures of the BDM approach in humans - surely we should not assume monkeys have better understanding of the task than humans do without testing it.

For (2), I encourage the authors to think about their result showing that monkeys preferences change a lot on a trial by trial basis depending on whether they won or lost the most recent gamble. This effect is highly suspicious, and ought to make them wary of a strictly value-based

interpretation. Specifically, my hunch is the monkeys assume the environment has some trial-to-trial fluctuation in its payoff structure, and the monkeys are trying to exploit that. Indeed, one could imagine a different research team with different theoretical orientation but the same data, writing that dopamine neurons encode strategic bias, not subjective value, because surely subjective value doesn't vary that quickly or strongly on a trial to trial basis.

For (3), let us consider, for example, the idea that monkeys may be highly risk-seeking, as this group has persuasively shown in the past. Such monkeys then are motivated to overbid low subjectively valued offers, because it is the overly valued risky choice. Then the subjective value of the option as inferred from the BDM procedure will be systematically greater than the true subjective value of the item. They will also underbid highly value offers, further complicating the matter. (The Bull paper cited above goes into the difficulties non-neutral risk attitudes cause for BDM, which are quite complex).

For (4) and (5) consider that monkeys are far from idealized subjects - they often get distracted, choose randomly, develop side biases, etc. The authors' method cannot distinguish true changes in subjective value from these other factors, and so, the dopamine neuron effects may simply be a proxy for these other factors, which are not value-related.

MINOR POINTS

Im just going to quote some sentences the authors may want to rethink, but there are many just as bad as these throughout the text:

"BDM bids are known to reflect the agent's true internal valuation, as inaccurate bidding results in suboptimal reward ('incentive compatibility')" This is a fishy statement - there are a whole bunch of cases in which monkeys make choices that result in suboptimal rewards, including many by this team. (E.g., risk-seeking). So if the logic of the argument requires the animals always choose optimally, it's a flawed argument.

"Support Vector Regression demonstrated accurate prediction of the animal's bids by as few as twenty dopamine neurons, demonstrating the validity of the dopamine code for internal reward value." Why does this demonstrate validity?

"...but we do not know whether these neuronal signals simply reflect the external stimulus indicating the subjective reward value or represent the internal subjective value elicited by the stimulus." This feels deliberately un scholarly - there are dozens of studies showing it's subjective,

and not a single one showing it's the external stimulus. Do the authors find this existing work unpersuasive for some reason?

"The second-price nature of BDM prevents incorrect bidding" - this is not true.

"But inferring subjective value from the typically tested binary choices is very limited; an agent can choose only the higher or the lower valued option in a binary fashion." not really true, since we can average over trials.

Reviewer #3 (Remarks to the Author):

Key results:

In this study, the authors asked whether dopamine neurons respond to external value, or internally generated, subjective value. To do this, they trained monkeys on an auction-like task to make bids on visual stimuli for juice rewards. They made their bids using a lever and their bids were compared to a computer bid. Three reward magnitudes were used. Neural recordings of dopamine (DA) neurons in the ventral midbrain were recorded during the task. Neurons responded to reward magnitude, unsurprisingly. However, some DA neurons responded to the bid size, independent of stimulus reward magnitude, which suggests that phasic DA can encode internally generated, subjective value estimates. The authors also used decoding to demonstrate the DA responses predicted the upcoming bids. Overall, this finding is useful for the field and advances our understanding of the role of dopamine in reward and value representation.

Validity:

The conclusions from the data seem valid. The writing could use some improvement for clarity (see below), but the interpretation of the data and statistical analyses seem sound.

Significance:

I think the authors could expand on the significance in the Introduction and Discussion. There was a heavy emphasis on describing BDM in the Introduction and rehashing the results in the Discussion multiple times. For example, in the discussion, the paragraph regarding decoding could be minimized, as I do not think the decoding really represents the strength of the main finding and anyway, and thus deserves less real estate. Both sections could reference the senior author's work more thoughtfully, as well as other work regarding the coding of subjective value (e.g. Kobayashi & Hsu 2019; O'Doherty, Buchanan, Seymour, Dolan 2006 ; Castrellon et al., 2019. to name a few).

Suggestions:

The task is explained well, but it remains unclear to me how the monkeys are encouraged to vary their bid size. From what I understand, they get juice or water on every trial, if they guess higher than the computer, they get the juice reward and the water that is the difference between the bid and the remainder. If they guess lower, then they get 1.2 ml of water but no juice. I'm definitely misunderstanding something, so could the authors please add a comment regarding why the monkeys don't just guess the highest they can each trial to get the maximum possible juice, or, the lowest every trial, and always get a water reward? Also, how were the juice values (volumes) for each monkey determined?

It seems that given the role of DA in learning, that this task, which appears to use the same 3 overlearned stimuli, might lead to results that would be different, had the monkeys been asked to learn 3 new stimuli each block. Could the authors please comment on this?

The writing around Figure 2 is a bit confusing and seems to suggest what is only shown by Figures 3 and 4. For example "The example neuron in figure 2a,b, exhibited increased activity with increase of both reward magnitude and bid in response to the onset of the fractal indicating the juice amount." Figure 2ab shows rasters that show higher rates for the largest magnitude image, and (b) is a little misleading in that bid is not broken out by image, and bid size and reward magnitude are correlated. I get that this comes later, but the conclusion regarding responsiveness to bid size independent of reward magnitude can't come without the support, which seems to arise first in figures 3 and 4.

There appears to be a timing difference in the neural responses to bid size (e.g. Fig 2b, 3d, 3e). Could the authors please discuss the reaction times for the different bid sizes? This seems critical for multiple reasons (1) to support the lack of motor response otherwise analyzed (2) understanding the timing of the predictive activity discussed later during the decoding analyses. It would be useful to see plots of the RTs for both bid size and image for both animals, and for the authors to analyze possible correlations between these and neural activity patterns, to rule out RT effects on the neural responses.

Regarding the decoding, how were time windows selected? How were reaction times controlled for in this analysis?

I really don't think there needs to be so much emphasis on BDM in the intro or discussion. It is a tool to create an auction-like environment with subjective values to analyze, but beyond that I think the time could be used to discuss the impact of the result rather than having the constant reminder of that acronym, without thoroughly tying the finding into other work that has used that framework. Would this experiment only have worked using BDM? It makes the experiment seem less generalizable when written this way.

Perhaps I missed it--is there a legend for the Table somewhere?

Can you please use the same Y axis ranges for the comparison you are trying to make in Figure 3c/d/e? If the emphasis is across monkeys, then please match those ranges, if the emphasis is within a monkey, then please match those. This figure is currently very misleading with the axes all different. Also the conclusion does not really match for Monkey U, as it seems the neurons respond

mostly to high bid + high reward, not just bid size, as for mid and low stimuli, the firing rates don't change as a function of bid size.

Overall the writing was straightforward and clear. I think the manuscript could be improved with a stronger nod to past work in various model systems and humans in the Introduction and a Discussion that helps the reader understand more clearly how this finding either updates, rejects, modifies, or supports our current understanding of the role of DA neurons in representation of value.

We would like to thank the reviewers for their insightful comments and suggestions. They have
greatly improved the clarity of the writing and the cohesiveness of the manuscript.

We have responded to each point below and have specifically addressed each of the concerns in
the manuscript. Note that gray highlighting in this document reflects specific points from the
reviewer that we responded to. In the manuscript document, gray highlighting represents changes
or additions to the text.

Reviewer #1 (Remarks to the Author):

In this manuscript, the authors make use of a well-defined and heavily validated behavioral task
couched in economic theory – the Becker-DeGroot-Marschak second-price auction (BDM) - to
identify the neural correlates (dopamine (DA) neuronal firing) of subjective value estimates.
Previous definitions of value and/or preference have often relied on modeling decisions across
multiple trials to estimate value. Here, the authors put forth a method that requires no modeling
or estimation; the methodology and data demonstrate that immediate value can be quantified in
the firing patterns of DA neurons in the minds of those making value estimates.

The findings are highly significant: DA neurons encode, not the discrete external-world values of
cues assigned by investigators, but the internal subjective values that individuals themselves
experience on a moment-by-moment basis.

The work quantitatively associates the notion of value, often assigned by investigators designing
and carrying out neuroeconomic paradigms, with subjects' own internal estimates of value, and
therefore brings the field closer to understanding the neural correlates of preference. The authors
may consider mentioning preference explicitly in order to broaden the interpretation of their
findings.

REFERENCE 1: My one significant comment about the data and the figures; There appears to be
activity prior to the grey shaded region in the figures. It is very interesting but I do not feel it is
adequately described or addressed. There are some comments below that pertain to this and the
use and definition of the gray shaded region in the figs.

AUTHORS' REPLY 1: 'activity prior to the grey shaded region in the figures'

Thank you for spotting the absence of a necessary explanation. We have now described in detail
that there are two dopamine response components, an early response that is attentional and not
relevant to our study, and a second response component that serves as the value prediction error;
this value signal was used for our analyses. We demonstrated these two dopamine response
components in our previous work (figures 2d, 4b, c, S6, S9 in Lak et al. 2014; figures 3a, 4a, 5f,
S3b in Stauffer et al. 2014; for review, see Schultz 2016). This is now described and explained
throughout the Results section (lines 162-198, 228-245) and in the legends for figs 2, 3, 4, S3.

AUTHORS' REPLY 2: 'use and definition of the gray shaded region'

We have also more carefully defined gray shaded region in the methods and figure legends.

REFERENCE 1: As for more potential analysis of the currently available data: the authors mention a
correlate with win/lose bidding. Does winning or losing a bid increase or decrease subsequent

bidding? I think finding a way to look at bidding across trials would be very interesting. Note: I
do see that there is an effect of previous bid and result (Fig 1e), but how might one interpret what
is going on across multiple trials? Do multiple losses occur in a row, multiple wins; does this
influence bidding?)

AUTHORS' REPLY 1: 'Does winning or losing a bid increase or decrease subsequent bidding?'
Wins and losses were coded as 1 and 0, respectively, in the mixed effects model presented in
figure 1e. Because we saw a positive association between win/lose and bidding (Fig. 1e), this
means wins result in higher bidding and losses result in lower bidding on the subsequent trial.
This is also reflected in the additional analysis described below.

AUTHORS' REPLY 2: 'Do multiple losses occur in a row, multiple wins; does this influence
bidding?'

We have added an analysis that incorporates consecutive wins and losses for the same reward
magnitude. The new result is interesting and suggests that sequential wins/losses affect the
monkey's value uniquely. Text has been added to the methods and results describing the analysis
and a new figure has been added to figure S2.

REFEREE 1: I think the discussion section could be improved; there are important points therein
but it feels a little disorganized.

AUTHORS' REPLY: 'the discussion section could be improved'
We have re-written and re-organized a number of paragraphs in the Discussion (as detailed below
and marked in gray in the revised text) and hope that will resolve the problem.

REFEREE 1: The authors might consider future studies documenting DA responses to
naturalistic affective stimuli, including images of predators or social stimuli. Typically these
stimuli are valued, but are often not paired with reward in a learning context.

AUTHORS' REPLY: 'DA responses to naturalistic affective stimuli'
We have added text to the discussion section to echo the need for this type of work as it relates
affective stimuli.

The following is a list of edits and in some cases stylistic suggestions.

REFEREE 1: Figures
A general comment related to task diagram and subsequent plots of neural activity. Would you
please explicitly put task timing info on the figure in 1a? I don't see that in the text or figure
legend. This is related to second point: I think all your plots are aligned to fractal on time, but I
only found that in 1 obvious place, the legend of fig 2. You might explicitly put on each t=0 line
the text "fractal on" or "cue offered" or something of the like.

AUTHORS' REPLY 1: 'task timing info on the figure in 1a'
Task timing has been added to figure S1 and we have to the description of the task and relevant
task events in the text and legends.

AUTHORS' REPLY 2: 't=0 line the text "fractal on" or "cue offered" or something of the like'

Arrows and text have been added to each figure to indicate the cue.

REFEREE 1: What is the gray shaded region in the neural plots? It says “post-event time period”
but what is that and how was it defined? I assume this is your ROI, but how was it selected? It
looks like there is information encoded in the period before that grey shaded region in Figs 2c, d
and 3c-e. When is the fractal cue turned off / or bid line offered to the monkey?

AUTHORS’ REPLY 1: ‘What is the gray shaded region in the neural plots. It says “post-event
time period” but what is that and how was it defined?’

The gray shaded area is the analysis time window. We have added text to the methods section
and to the figure 2 and 3 legends to more clearly describe the analysis window.

AUTHORS’ REPLY 2: ‘period before that grey shaded region in Figs 2c, d and 3c-e’

The issue is the same two-component dopamine response as explained above by the early
response that is attentional (and not very relevant), and a second response component that is the
true value prediction error and constitutes the data for this study. We had demonstrated these two
dopamine response components in our previous work (figures 2d, 4b, c, S6, S9 in Lak et al. 2014;
figures 3a, 4a, 5f, S3b in Stauffer et al. 2014; for review, see Schultz 2016). This is now
described and explained throughout the Results section (lines 162-198, 228-245) and in the
legends for figs 2, 3, 4, S3.

REFEREE 1: The t=0 line is very faint in figs 2a; missing in 2c and d; faint in 3a; missing in 3 c-
3; missing in Fig 4a-c. You may or may not want to include it in all figs.

AUTHORS’ REPLY: ‘The t=0 line is very faint...missing...’

We have added black lines to all plots to indicate t = 0 and descriptors to indicate the event
(‘Rew. Cue’).

REFEREE 1: In summary, please consider how best to display task timing info in Fig 1a and how
to highlight the relevant epochs in the remaining figs (and clarify what the gray region is and
how/why it was selected).

AUTHORS’ REPLY: ‘how best to display task timing info’

Please see our replies just above for details.

REFEREE 1: Supplemental Figures

For the indicated prediction error and task events shown in S1; does the indicated event refer to
the start and subsequent epoch? Is the reward cue a prediction error event (and not task event)?
Perhaps clarify these definitions? And consider using visual braces to indicate the epochs of
interest.

AUTHORS’ REPLY 1: ‘Is the reward cue a prediction error event (and not task event)?’

Yes, it is a prediction error, relative to the preceding trial-start cue. This is now stated in Results,
subtitle ‘Dopamine signal reflects trial-by-trial changes in subjective value’, first and second
paragraphs (lines 167 and 173). Text has also been added to Figure S1 legend for further
clarification.

AUTHORS' REPLY 2: 'does the indicated event refer to the start and subsequent epoch?'...
'consider using visual braces to indicate the epochs of interest'
Timing information and text have been added to fig S1 to more clearly indicate epochs of
interest.

REFEREE1 : Line 67

Why not refer to preference here? Stimuli and outcomes we "like" - independent of secondary
reinforcers - seem more akin to the internal values you are investigating than the mentioned
references about musical tunes and imagined speech. Or say: "Neuronal signals for internal
processes exist. For example..." Is there not a literature on neural signals related to viewing and
rating fine art, or attractive faces etc?

AUTHORS' REPLY 1: 'Why not refer to preference here?'

We have added that value is inferred from choice preferences in line 59 (strictly spoken,
preference and choice is not the same thing although it is sometimes used interchangeably by
economists; if we want to use it, we would need to talk about revealed preference when studying
monkey choices, as opposed to preferences stated ahead of choice that are only feasible in
humans; the specification would get complicated, and therefore we like to keep using the word
'choice' and only mention preference occasionally).

AUTHORS' REPLY 2: 'neural signals related to viewing and rating fine art, or attractive faces
etc'

We have added citation of two BDM studies to the Introduction (line 79).

REFEREE 1: Line 82

Missing a word here, probably "us" or "one"

AUTHORS' REPLY: 'Missing a word'

We changed this together with the revision of the penultimate Introduction paragraph (lines 84-
85).

REFEREE 1: Line 91

The phrasing "externally imposed" seems somewhat awkward. Alternative: "... than values
defined by external task constraints"?

AUTHORS' REPLY: "'externally imposed" seems somewhat awkward'

Thank you, we changed this as suggested (end of Introduction, line 94).

REFEREE 1: Line 112

I don't believe there is text in the ms to describe figure 1b (but do see it in the figure legends).
Please briefly describe Fig 1b in the ms text.

AUTHORS' REPLY: 'I don't believe there is text in the ms to describe figure 1b'

Thank you for spotting this. We have added citation of figure 1b to the revised description of the
BDM mechanism in lines 113-121.

REFEREE 1: Line 119

Are you saying that the SD of bidding should be not statistically different for the 3 volumes – if
driven by internal value? I don't follow. How is the variance an indicator of internal bidding
(fidelity)? There are likely motivational effects for the three different reward volumes, No? I do
see your analysis of coherence (supplemental 2b), but the distributions on the right in the small
panel of Fig 1c look different. Do these curves look different because of bidding over time? It
looks like it comes up with the high and mid magnitude cues.

AUTHORS' REPLY 1: 'How is the variance an indicator of internal bidding?'
MAD (Figure S2a) is not meant to indicate internal value, rather, it is meant to demonstrate how
variable bidding is from trial to trial. It also shows that bids vary to a similar degree across the
three levels. This can be interpreted in one of two ways: either the bids vary randomly or by some
unknown/unmeasured process, or the bids vary systematically with the internal value. If the bids
vary systematically with value, we can assume that the value would change in the same way for
all three reward magnitudes (as opposed to stochastically). To address this, we measured the bid
coherence. Figure S2b shows that the bids do exhibit a significant level of coherence, suggesting
they vary somewhat systematically. It is critical to note that this measure of coherence is likely a
lower limit estimation since the reward magnitudes were not presented sequentially or
simultaneously. Additionally, the relatively low coherence might suggest that there were other
immeasurable phenomena that did contribute to bid variability. In spite of this, dopamine neurons
still encoded the internal value remarkably well. We have modified the text on line 129 and 130
to better clarify this point.

AUTHORS' REPLY 2: 'Do these curves look different because of bidding over time? It looks
like it comes up with the high and mid magnitude cues.'
Fig 1c represents a single session and, although representative with of bid variance and
coherence, the shapes of the bid distributions did vary from day-to-day. However, the way that
bids change over time might differ among reward magnitudes due to satiety (as suggested by
figure 1e). That is, as the animal becomes more sated generally (total liquid intake), the value of
water likely decreases and the value of juice may increases or stay the same. This would certainly
lead to an upward non-stationarity (as depicted in fig1c), which would result in a more
platykurtic distribution. Although this analysis is beyond the scope of the current study, we hope
to explore this in future work.

REFEREE 1: I believe your demonstration that bid variability results from changes in internally
generated subjective values, but how this relates to coherence, or is verified by bid coherence, is
unclear to me.

AUTHORS' REPLY 2: 'how this relates to coherence'.
Coherence in this case is defined as the degree to which bids for the three reward magnitudes
'move together' over time. Coherence suggests that even though there are fluctuations in bidding,
there is an underlying internal value that is driving these changes across all three levels. We have
added text from, lines 129 to 136 to better describe this metric and interpretation of the analysis.

REFEREE 1: Line 131
You say: "factors that significantly affected the animal's bids...". But what about the bids? Their
quantitative value?

AUTHORS' REPLY: 'But what about the bids? Their quantitative value?'

Yes. Bids are the dependent variable in this case. These regression analyses were meant to

answer the general question, 'What drives changes in bidding?'. The lasso regression was used to

eliminate predictor variables that did not contribute to the variance of the outcome variable

(bids). Text was added to this section (now found on lines 138 and 139) to better describe how

the lasso regression was utilized.

REFEREE 1: Line 132

The win/lose effect is intriguing! What is the direction of the effect? Is this shown or addressed

somewhere (I have overlooked)? Does winning or losing a bid increase or decrease subsequent

bidding? Either for the same cue or any cue?

AUTHORS' REPLY: 'What is the direction of the effect?...Does winning or losing a bid

increase or decrease subsequent bidding?'

The mixed effects model shows that there is a positive relationship between win/lose and bids.

This suggests that wins result in higher bidding on the subsequent trial and losses result in lower

bidding. We have added text to the results section to describe this finding more clearly.

REFEREE 1: Line 152

"... cue for water payout..." how do you know it is this cue, in particular? From the task diagram

in Fig 1a, it looks like there is a lot of available info at that point in the trial.

AUTHORS' REPLY: 'how do you know it is this cue, in particular?'

The cue for the water payout as indicated by the competing bid is one of only two points in the

task where there is a prediction error (the other being the cue for the reward, the fractal). This is

because the animal does not know where the randomly-generated competing bid will appear.

Therefore, when the competing (computer-generated) bid appears on screen, this generates a

prediction error and we would expect DA neurons to exhibit a graded response. When identifying

the number of DA neurons that exhibited a graded response, we used this cue and the fractal

reward cue. We have added more detail to figure S2 and added additional text to the figure

legend to better describe task events and prediction errors.

REFEREE 1: Line 160

does this neuron show suppression for medium and low reward magnitudes – during the grey

shaded region? I am confused about the gray shading. It looks like DA-neuron activity encoding

for small and medium peaks before the large reward cue peaks? This is interesting; do you have

thoughts about it?

REFEREE 1: Suppression during gray shaded time period?

Why the early peak for mid and low volumes and later peak for max volume?

REFEREE 1: Fig 2d

note early peak for low bids and later peak for high bids - why? Have you done some sort of

temporal analysis to identify when DA neurons encode the internal bid estimates?

REFEREE 1: Line 191; Fig 3d, 3e

Still - lower bids show earlier spiking! Especially in mk V. What does this mean?

**AUTHORS' COMMON REPLY TO THE FOUR REFEREE COMMENTS ABOVE** : The issue
is the same two-component dopamine response as explained above and explained by the early
response that is attentional (and not very relevant), and a second response component that is the
true value prediction error and constitutes the data for this study. We had demonstrated these two
dopamine response components in our previous work (figures 2d, 4b, c, S6, S9 in Lak et al. 2014;
figures 3a, 4a, 5f, S3b in Stauffer et al. 2014; for review, see Schultz 2016). This is now
described and explained throughout the Results section (lines 162-198, 228-245) and in the
legends for figs 2, 3, 4, S3.

Peter M Kaskan

**AUTHORS' REPLY:** thank you for the nice and constructive comments!

Reviewer #2 (Remarks to the Author):

This paper used an economic method, the BDM auction, to assess subjective trial by trial values
of options, to monkeys. The authors then assess values of offers in a computerized task and ask
whether dopamine responses correspond more closely to subjective or to objective value. They
find the answer is subjective value.

**REFEREE 2:** These results are not particularly surprising or important. There have been dozens
of studies asking whether neurons or hemodynamic responses more faithfully encode subjective
or objective values, going back over 20 years. In every single case, the answer was always
exactly the same: the brain encodes subjective values, not the objective values. The authors here
find exactly the same thing. So there's nothing surprising about these results, nor do they change
existing theories, nor do they provide needed support for any proposed, or even generally
accepted but unsubstantiated theories.

**AUTHORS' REPLY:** 'There have been dozens of studies asking whether neurons... encode
subjective or objective values...'

Yes, there are many studies that showed subjective value coding, but what we show with BDM is
something different: the dopamine neurons code the internal subjective value rather than the
subjectively weighted objective value shown in these other studies. We have explained and
emphasized this point better in the rewritten three paragraphs in the Introduction that replace the
two original paragraphs (lines 59-86).

**REFEREE 2:** The interesting thing about the paper is the use of the BDM task, but, of course,
that is not novel here. The authors have already written that method/result up in a J Neuroscience
paper last year.

There's another, deeper problem with that the paper that the authors seems to be unaware of, so I
will mention it here. The BDM only works given certain assumptions that are almost certain to be
violated in this case (see, e.g., Plott and Zeiler, 2005). These include (1) the idea that the subjects
fully understand the structure of the BDM auction (Chou et al., 2009), (2) that they fully trust the
experimenter and the stability of the task, and (3) that they are risk-neutral, (4), that subjects have

access to their own internal values and (5) that variability in performance is due to variability in
preference not things like task engagement, distraction, etc.

Indeed, the BDM, while great in theory, more often than not fails in practice with humans - and
monkeys are sure to be worse (for a good recent review of this topic, see Bull, Courty, Doyon,
and Rondeau, 2019).

For (1), I encourage the authors to read Cason and Plott (2014), which deals with this issue, and
demonstrates notable failures of the BDM approach in humans - surely we should not assume
monkeys have better understanding of the task than humans do without testing it.

AUTHORS' REPLY 1: '(1) subjects fully understand the structure of the BDM auction' and 'not
assume monkeys have better understanding of the task than humans do':

Yes, that is correct, and we surely don't know whether the monkeys understood the structure of
the BDM; also the conditions in which BDM is performed differs hugely between humans and
our monkeys who performed over several months and showed good consistency, as shown in
figure 1c, d, e and our previous paper (Al-Mohammed & Schultz 2022). These points are now
incorporated into the revised paragraph, together with citing the very fitting Cason & Plott 2014
paper (thank you). The revised part is found in the second Discussion paragraph (lines 336-346).

REFEREE 2: For (2), I encourage the authors to think about their result showing that monkeys
preferences change a lot on a trial by trial basis depending on whether they won or lost the most
recent gamble. This effect is highly suspicious, and ought to make them wary of a strictly value-
based interpretation. Specifically, my hunch is the monkeys assume the environment has some
trial-to-trial fluctuation in its payoff structure, and the monkeys are trying to exploit that. Indeed,
one could imagine a different research team with different theoretical orientation but the same
data, writing that dopamine neurons encode strategic bias, not subjective value, because surely
subjective value doesn't vary that quickly or strongly on a trial to trial basis.

AUTHORS' REPLY: '(2) preferences change a lot on a trial by trial basis depending on whether
they won or lost the most recent gamble'

The previous trial outcome (win/lose) has a limited effect as shown by the regressor strength in
fig 1e (far right), which may indicate instantaneous adaptation and possibly other challenges to
accurate BDM bidding mentioned by Plott & Zeiler 2005 (thank you for the reference). We have
inserted two sentences describing this in the third Discussion paragraph (lines 333-336).

REFEREE 2: For (3), let us consider, for example, the idea that monkeys may be highly risk-
seeking, as this group has persuasively shown in the past. Such monkeys then are motivated to
overbid low subjectively valued offers, because it is the overly valued risky choice. Then the
subjective value of the option as inferred from the BDM procedure will be systematically greater
than the true subjective value of the item. They will also underbid highly value offers, further
complicating the matter. (The Bull paper cited above goes into the difficulties non-neutral risk
attitudes cause for BDM, which are quite complex).

AUTHORS' REPLY: '(3) monkeys may be highly risk-seeking'

Yes, our previous data show that monkeys are risk-seeking for small rewards (<0.5 ml of juice or
water) and risk-avoiders for larger rewards (>0.7 ml) (e.g. Stauffer et al. 2014). In agreement

with the referee's comment, the bids would not be linear on reward amount, but should in their
slope incorporate the convexity-linearity-concavity suggested by the risk attitude. We have
therefore added a sentence in lines 149-151 saying, "Note that the use of only three reward
magnitudes has reduced bidding nonlinearities reflecting the animal's risk attitude, as BDM
bidding is inherently risky due to the randomly bidding computer opponent."

REFEREE 2: For (4) and (5) consider that monkeys are far from idealized subjects - they often
get distracted, choose randomly, develop side biases, etc. The authors' method cannot distinguish
true changes in subjective value from these other factors, and so, the dopamine neuron effects
may simply be a proxy for these other factors, which are not value-related.

AUTHORS' REPLY 1: '(4) and (5)...get distracted, choose randomly'

The systematic and consistent ranking of bids shown in figure 1c,d and e argue against distracted
and random choice. We have added a conclusion sentence in lines 154-155 that says 'The
systematic and consistent bidding over several months argues against distracted and random
bidding behavior of these monkeys.'

AUTHORS' REPLY 2: '(4) and (5)...develop side biases'

The joystick-lever and the bid cursor were both always located on the right-hand side of the
animal and bids were made by moving the cursor up or down in order to prevent side bias. We
now mention this explicitly in Methods, end of paragraph subtitled 'Experimental setup' (lines
428-433).

AUTHORS' REPLY 3: '(4) and (5)...dopamine neuron effects may simply be a proxy for these
other factors, which are not value-related'

The systematic and consistent increase of dopamine responses with reward amounts (figure 2c)
and with bids (figure 2d) argue against factors other than some form of value. This is now more
clearly stated in Discussion, first paragraph, lines 310-314.

MINOR POINTS

Im just going to quote some sentences the authors may want to rethink, but there are many just as
bad as these throughout the text:

REFEREE 2: "BDM bids are known to reflect the agent's true internal valuation, as inaccurate
bidding results in suboptimal reward ('incentive compatibility')." This is a fishy statement - there
are a whole bunch of cases in which monkeys make choices that result in suboptimal rewards,
including many by this team. (E.g., risk-seeking). So if the logic of the argument requires the
animals always choose optimally, it's a flawed argument.

AUTHORS' REPLY: 'monkeys make choices that result in suboptimal rewards'

Yes, the way we had formulate it can be seen as contentious and possibly inaccurate. We have
now softened this sentence in the Abstract to 'The BDM mechanism encourages agents to reveal
their true internal valuation, as inaccurate bidding results in suboptimal reward ('incentive
compatibility').'

REFEREE 2: "Support Vector Regression demonstrated accurate prediction of the animal's bids
by as few as twenty dopamine neurons, demonstrating the validity of the dopamine code for
internal reward value." Why does this demonstrate validity?

AUTHORS' REPLY: 'Why does this demonstrate validity?'

Yes, 'validity' is maybe too strong, but the SVR is usually seen as confirmation of a code
(including a neuronal code). We have therefore changed this part of the sentence in the Abstract
to '...confirming a neuronal code for internal subjective reward value'.

REFEREE 2: "...but we do not know whether these neuronal signals simply reflect the external
stimulus indicating the subjective reward value or represent the internal subjective value elicited
by the stimulus." This feels deliberately unscholarly - there are dozens of studies showing it's
subjective, and not a single one showing it's the external stimulus. Do the authors find this
existing work unpersuasive for some reason?

AUTHORS' REPLY: 'This feels deliberately unscholarly'

Yes, the statement may be perceived as unscholarly (but not deliberately). We had not clearly
made the distinction between internal subjective value revealed by BDM and subjective
weighting of objective reward amounts described in previous neurophysiological studies. As
stated above, the issue has now been addressed with the revision of these paragraphs in the
Introduction (lines 59-88).

REFEREE 2: "The second-price nature of BDM prevents incorrect bidding" - this is not true.

AUTHORS' REPLY: 'this is not true'

Yes, while the statement in itself should be in principle correct, what happens often in BDM
makes this message untrue: humans often don't understand the BDM, and thus their bidding is
wrong, and our sentence can be seen as being invalid. We have adjusted these text portions in the
Discussion paragraph on lines 336-346 as stated in our reply above to issue (1) of this referee.

REFEREE 2: "But inferring subjective value from the typically tested binary choices is very
limited; an agent can choose only the higher or the lower valued option in a binary fashion." not
really true, since we can average over trials.

AUTHORS' REPLY: 'not really true, since we can average over trials'

To transform this into a true statement requires more explanation. The issue is treated in the
subsequent three revised paragraphs. We have therefore removed this whole sentence (on lines
58-60 in the original version).

Reviewer #3 (Remarks to the Author):

Key results:

In this study, the authors asked whether dopamine neurons respond to external value, or
internally generated, subjective value. To do this, they trained monkeys on an auction-like task to
make bids on visual stimuli for juice rewards. They made their bids using a lever and their bids
were compared to a computer bid. Three reward magnitudes were used. Neural recordings of

dopamine (DA) neurons in the ventral midbrain were recoded during the task. Neurons responded
to reward magnitude, unsurprisingly. However, some DA neurons responded to the bid size,
independent of stimulus reward magnitude, which suggests that phasic DA can encode internally
generated, subjective value estimates. The authors also used decoding to demonstrate the DA
responses predicted the upcoming bids. Overall, this finding is useful for the field and advances
our understanding of the role of dopamine in reward and value representation.

Validity:

The conclusions from the data seem valid. The writing could use some improvement for clarity
(see below), but the interpretation of the data and statistical analyses seem sound.

Significance:

REFEREE 3: I think the authors could expand on the significance in the Introduction and
Discussion. There was a heavy emphasis on describing BDM in the Introduction and rehashing
the results in the Discussion multiple times. For example, in the discussion, the paragraph
regarding decoding could be minimized, as I do not think the decoding really represents the
strength of the main finding and anyway, and thus deserves less real estate. Both sections could
reference the senior author's work more thoughtfully, as well as other work regarding the coding
of subjective value (e.g. Kobayashi & Hsu 2019; O'Doherty, Buchanan, Seymour, Dolan 2006 ;
Castrellon et al., 2019. to name a few).

AUTHORS' REPLY: 'expand on the significance in the Introduction and Discussion'
We have amended the discussion and introduction to elaborate on the significance of the task and
the findings.

Suggestions:

REFEREE 3: The task is explained well, but it remains unclear to me how the monkeys are
encouraged to vary their bid size. From what I understand, they get juice or water on every trial,
if they guess higher than the computer, they get the juice reward and the water that is the
difference between the bid and the remainder. If they guess lower, then they get 1.2 ml of water
but no juice. I'm definitely misunderstanding something, so could the authors please add a
comment regarding why the monkeys don't just guess the highest they can each trial to get the
maximum possible juice, or, the lowest every trial, and always get a water reward? Also, how
were the juice values (volumes) for each monkey determined?

AUTHORS' REPLY: 'why the monkeys don't just guess the highest they can get in each trial':
The reason why the monkeys don't just guess the highest value is that they pay the price of losing
water according to the computer bid they exceeded when winning the auction: If they guessed too
high, and won the auction, the computer bid may have also been too high, and the animal would
lose too much water reward, thus preventing over-bidding. This is the 'incentive compatible'
nature and principle of a second-price auction that elicits the bid according to the price the agent
is willing to pay (= the true internal value). We have explained and emphasized this point better
in the revised part of the Introduction (lines 76-88) and the revised beginning of the Results
section (lines 114-121).

REFEREE 3: It seems that given the role of DA in learning, that this task, which appears to use
the same 3 overlearned stimuli, might lead to results that would be different, had the monkeys

been asked to learn 3 new stimuli each block. Could the authors please comment on this?

AUTHORS' REPLY: 'same 3 overlearned stimuli' and 'results that would be different, had the
monkeys been asked to learn 3 new stimuli each block'

The same two monkeys showed similarly consistent bidding behavior with five different fractals
for five juice volumes, which had been reported in the behavioral part of the study (Al-
Mohammad & Schultz 2022). We now state this in an added short paragraph to the Methods
section (lines 519-523).

REFEREE 3: The writing around Figure 2 is a bit confusing and seems to suggest what is only
shown by Figures 3 and 4. For example "The example neuron in figure2a,b, exhibited increased
activity with increase of both reward magnitude and bid in response to the onset of the fractal
indicating the juice amount." Figure 2ab shows rasters that show higher rates for the largest
magnitude image, and (b) is a little misleading in that bid is not broken out by image, and bid size
and reward magnitude are correlated. I get that this comes later, but the conclusion regarding
responsiveness to bid size independent of reward magnitude can't come without the support,
which seems to arise first in figures 3 and 4.

AUTHORS' REPLY: 'The writing around Figure 2 is a bit confusing and seems to suggest what
is only shown by Figures 3 and 4'

Yes, the data shown introduce the dopamine responses without distinguishing between bid and
reward magnitude and, as the referee rightly says, that distinction comes only later. While we
cannot at this point jump to the later analysis before describing more details of the current result,
we have nevertheless added a short 'preview' sentence saying that later analyses showed that the
neuronal response increased with bid independently of reward magnitude (lines 204-205).

REFEREE 3: There appears to be a timing difference in the neural responses to bid size (e.g. Fig
2b, 3d, 3e). Could the authors please discuss the reaction times for the different bid sizes? This
seems critical for multiple reasons (1) to support the lack of motor response otherwise analyzed
(2) understanding the timing of the predictive activity discussed later during the decoding
analyses. It would be useful to see plots of the RTs for both bid size and image for both animals,
and for the authors to analyze possible correlations between these and neural activity patterns, to
rule out RT effects on the neural responses.

AUTHORS' REPLY 1: 'timing difference in the neural responses'

The issue is about the same two-component dopamine response as explained above in reply to
comments by Referee 1 and explained by the early response that is attentional (and not relevant),
and a second response component that is the true value prediction error and constitutes the data
for this study. We had demonstrated these two dopamine response components in our previous
work (figures 2d, 4b, c, S6, S9 in Lak et al. 2014; figures 3a, 4a, 5f, S3b in Stauffer et al. 2014;
for review, see Schultz 2016). This is now described and explained throughout the Results
section (lines 162-198, 228-245) and in the legends for figs 2, 3, 4, S3).

AUTHORS' REPLY 2: 'timing of the predictive activity discussed later during the decoding
analyses'

We used the dopamine responses from the second, value response component while neglecting
responses in the first, attentional response component. This is now stated in Results, under
subtitle ‘Dopamine responses decode future bids’, lines 274-277.

REFEREE 3: Regarding the decoding, how were time windows selected? How were reaction
567 times controlled for in this analysis?

AUTHORS’ REPLY: ‘time windows’:

Also in reply to Referee 1, we now provide this information in the Experimental procedures
section (lines 581-596). As mentioned above, we used the time windows for the second, value
response component of the dopamine neurons while neglecting responses in the first, attentional
response component. This is now stated in Results, under subtitle ‘Dopamine responses decode
future bids’. Similarly, the method for selecting the analysis time windows has been elaborated in
the methods

REFEREE 3: I really don’t think there needs to be so much emphasis on BDM in the intro or
discussion. It is a tool to create an auction-like environment with subjective values to analyze, but
beyond that I think the time could be used to discuss the impact of the result rather than having
the constant reminder of that acronym, without thoroughly tying the finding into other work that
has used that framework. Would this experiment only have worked using BDM? It makes the
experiment seem less generalizable when written this way.

AUTHORS’ REPLY: ‘don’t think there needs to be so much emphasis on BDM...Would this
experiment only have worked using BDM?’

The experiment would not have been interesting without having used the BDM, as also referee 1
above noted. The reason we used the BDM is that it is the only mechanism that reveals the true
internal subjective value on a moment-by-moment basis. Also following the comments of referee
1, we have revised the Introduction accordingly with three reformulated paragraphs (lines 59-86).

REFEREE 3: Perhaps I missed it--is there a legend for the Table somewhere?

AUTHORS’ REPLY: ‘is there a legend for the Table somewhere?’

A legend has been provided for Table S1 on lines 832 through 838.

REFEREE 3: Can you please use the same Y axis ranges for the comparison you are trying to
make in Figure 3c/d/e? If the emphasis is across monkeys, then please match those ranges, if the
emphasis is within a monkey, then please match those. This figure is currently very misleading
with the axes all different. Also the conclusion does not really match for Monkey U, as it seems
the neurons respond mostly to high bid + high reward, not just bid size, as for mid and low
stimuli, the firing rates don’t change as a function of bid size.

AUTHORS’ REPLY 1: ‘same Y axis ranges’: We have amended the Y axis to be the same for
each reward magnitude for both animals.

AUTHORS’ REPLY 2: ‘conclusion does not really match for Monkey U’

The discretization of the bids within each reward level is meant to serve as a rough visualization
of how the neuronal signals coincide with the bids. Due to individual differences in bid-encoding
(e.g., different animals and subsets of their neurons likely encode high and low rewards with
different dynamic ranges and different response slopes; see Dabney et al. 2020), when the bids
were discretized into evenly split bins, the average traces did not accurately reflect the final result
in figure 3f-h. We have now amended this figure so that bids are split into terciles and corrected
for skewness of the bid distribution. Also, we now describe and explain in much more detail the
variation of dopamine responses with bids independent from variations with reward magnitude
(second paragraph under subtitle ‘Dopamine neurons reflect subjective value (bids) irrespective
of reward magnitude’, lines 246-263).

REFERENCE 3: Overall the writing was straightforward and clear. I think the manuscript could be
improved with a stronger nod to past work in various model systems and humans in the
Introduction and a Discussion that helps the reader understand more clearly how this finding
either updates, rejects, modifies, or supports our current understanding of the role of DA neurons
in representation of value.

AUTHORS’ REPLY: ‘stronger nod to past work in various model systems and humans in the
Introduction and a Discussion’

We have now amended the introduction and discussion to include more citations of past work
and elaborate on the significance of the findings.

Reviewers' comments:

Reviewer #1 (Remarks to the Author):

Thank you for making the recommended changes.

Reviewer #2 (Remarks to the Author):

In the first round of review, I voted reject because the paper lacks novelty. In the revision, the authors have changed their narrative. Now, in addition to lacking novelty, the central message of the paper relies on a basic misunderstanding of economics. In addition, the authors have waved off my legitimate criticisms of their behavioral method without taking the critiques seriously.

This is a seriously flawed paper that will embarrass the authors if it is published in anything close to its present form.

The authors claim that there is a distinction between "internal subjective value" and "subjectively weighted objective value". This is nonsense. Those two terms refer to precisely the same thing.

I have been doing neuroeconomics for 20 years. I have been to the Society for Neuroeconomics meetings many times, I have seen hundreds of relevant talks, reviewed over a hundred papers and grants, dozens of them on the representation of value. I have never once seen anyone make this distinction. That is because there isn't one - these are simply two terms used to refer to the same thing.

It is clear the authors have never seen anyone make this distinction either - they do not cite any papers that make the distinction. If the authors believe they have developed a novel perspective on how value is computed, they ought to devote much more space to explaining this idea and justifying their counterintuitive claim; instead, they mention it only in passing, as if it is well-accepted in the field. It is not. I went back and read the other papers using the BDM method that the authors cite; these papers do not make this distinction either.

I think that the authors misunderstand what the BDM is. It is not a magical tool that gives you access to some secret form of value that no other method of elicitation can give you. It is simply a preference-based elicitation method, like all the others, but it is one that is fully incentive compatible. It also has major disadvantages that the authors seem to be unaware of, or else eager to gloss over. It is complicated and that makes it very finicky in practice, and that's with human subjects. I am 100% certain things are much worse with monkeys. It also requires many assumptions to be true for its results to be meaningful. As I stated in my earlier review, these assumptions are very unlikely to be true in this case. The authors do not address these critiques in their reply.

But even if they were true, and even if the monkeys understood the task, all the BDM can give you is "subjectively weighted objective value." That's it. Even the most ardent defenders of BDM would agree. That's what it was designed for.

Reviewer #3 (Remarks to the Author):

In my opinion, the authors have addressed the concerns of all three reviewers sufficiently and the new version of the manuscript has improved over the original version.

Replies to comments by reviewer 2 (second round)

REVIEWER: In the first round of review, I voted reject because the paper lacks novelty. In the revision, the authors have changed their narrative. Now, in addition to lacking novelty, the central message of the paper relies on a basic misunderstanding of economics. In addition, the authors have waved off my legitimate criticisms of their behavioral method without taking the critiques seriously.

AUTHORS' REPLY:

- *'Novelty' is demonstrated by the fact that nobody has ever tested the BDM auction-like mechanism on neurons, including reward neurons and dopamine neurons. The only BDM study ever on monkeys is our own behavioral study in the Journal of Neuroscience 2022, with extensive documentation by 16 figures. Testing the BDM on neurons is important and novel, as it presents an entirely different and more direct approach to subjective value than formal utility functions previously estimated in binary choices in our laboratory (Stauffer et al. 2014). Because of that more direct access to subjective value compared to standard binary choice, BDM is a widely used research tool in behavioral economics and mimics procedures used in practice in everyday transactions (e.g., Google advertising auctions, art auctions). We now emphasize novelty more clearly in the revised Introduction (near lines 67-69).*
- *'misunderstanding of economics': we have explained better the notion of subjective value in the revised Introduction (near lines 58-70).*

REVIEWER: This is a seriously flawed paper that will embarrass the authors if it is published in anything close to its present form.

AUTHORS' REPLY: ???

REVIEWER: The authors claim that there is a distinction between "internal subjective value" and "subjectively weighted objective value". This is nonsense. Those two terms refer to precisely the same thing.

AUTHORS' REPLY:

- *'precisely the same thing': the statement is wrong: the BDM is particularly useful as it does NOT assess subjectively weighted objective value; it does assess subjective value directly and thus does NOT rely on, or require a measure of, objective value. In BDM, the bidder states her value freely by making a bid for a good, and that stated bid is compared against a random computer bid: the auction is won if the stated bid equals or exceeds the computer bid, and is lost otherwise. By contrast, utility functions subjectively weigh objective, physical value, which is expressed in typical logarithmic, power, negative-exponential, quadratic or other mathematically defined functions, for many years (starting with Bernoulli 1738 and axiomatized by Von Neumann & Morgenstern 1944). We have now stated this more clearly in the revised Introduction (near lines 58-70) and Discussion (near lines 376-387) (our initial rebuttal letter provided essentially the same but more detailed description).*

REVIEWER: I have been doing neuroeconomics for 20 years. I have been to the Society for Neuroeconomics meetings many times, I have seen hundreds of relevant talks, reviewed over a hundred papers and grants, dozens of them on the representation of value. I have never once seen anyone make this distinction. That is because there isn't one - these are simply two terms used to refer to the same thing. It is clear the authors have never seen anyone make this distinction either - they do not cite any papers that make the distinction. If the authors believe they have developed a novel perspective on how value is computed, they ought to devote much more space to explaining this idea and justifying their counterintuitive claim; instead, they mention it only in passing, as if it is well-accepted in the field. It is not. I went back and read the other papers using the BDM method that the authors cite; these papers do not make this distinction either.

AUTHORS' REPLY:

• 'never once seen anyone make this distinction': *there is a big difference between direct subjective value assessment in an auction-like mechanism like the BDM and the estimation of a formal economic utility function that represents subjectively weighted objective value ($U(x)$, with U as mathematically defined utility function, and x as objective reward amount). The BDM does not rely on a mathematical function of objective value. There are different meanings of the words subjective value and utility: the general meaning relates to the satisfaction and benefit of the decision maker and contrasts with the specific meaning of formal mathematically defined subjective value and utility. We now emphasize these different meanings in the revised Introduction (near lines 64-69) (the length specification of the Journal prevents us from adding more extensive explanations).*

• 'have developed a novel perspective on how value is computed': *we are NOT claiming a novel perspective (just novel data, see above). The BDM has been known for 60 years, and we substantiate our non-invention by citing the original 1964 paper. We use the BDM precisely because it is such a well-established estimation mechanism for subjective value. In fact, the senior author (WS) spent many months discussing the advantages of the BDM for assessing subjective value in monkeys and their reward neurons with three renown Caltech economists: John Ledyard, Charlie Plott, and David Grether. Indeed, a specialist in mechanism design, John Ledyard suggested the experiment to us for demonstrating that dopamine reward responses signal subjective reward value in a more direct way than estimated formal utility as a function of objective value. The current study is the result of that initiative.*

REVIEWER: I think that the authors misunderstand what the BDM is. It is not a magical tool that gives you access to some secret form of value that no other method of elicitation can give you. It is simply a preference-based elicitation method, like all the others, but it is one that is fully incentive compatible. It also has major disadvantages that the authors seem to be unaware of, or else eager to gloss over. It is complicated and that makes it very finicky in practice, and that's with human subjects. I am 100% certain things are much worse with monkeys. It also requires many assumptions to be true for its results to be meaningful. As I stated in my earlier review, these assumptions are very unlikely to be true in this case. The authors do not address these critiques in their reply.

AUTHORS' REPLY: 'complicated and that makes it very finicky in practice' and 'I am 100% certain things are much worse with monkeys': *These assumptions by the referee derive from experience and evidence from human subjects, the only species in which the BDM was tested prior to our work. For exactly these*

reasons, we painstakingly characterized the BDM for several years (Al-Mohammad & Schultz 2022) in the same two monkeys before we used them for the current neurophysiological study (see the 16 figures in our JN 2022 paper). Each animal was tested in tens of thousands of trials in several preliminary and intermediate BDM versions until reliable performance was obtained that correlated well with conventional binary choices that served as reference. By contrast, humans are usually not well trained on the BDM, and their performance is therefore unreliable. These monkeys were veritable experts on this task. We now emphasize this point in the revised Abstract (near lines 41-42), Introduction (near lines 89-102), Results (near lines 106-108), Discussion (near lines 349-353) and Experimental Procedures (near lines 476-477).

REVIEWER: But even if they were true, and even if the monkeys understood the task, all the BDM can give you is "subjectively weighted objective value." That's it. Even the most ardent defenders of BDM would agree. That's what it was designed for.

AUTHORS' REPLY: *'all the BDM can give you is "subjectively weighted objective value"'. We have dealt with the issue as explained in detail above.*

REVIEWER COMMENTS

Reviewer #4 (Remarks to the Author):

This study by Hill and colleagues examined whether the activity of dopamine neurons correlates with trial-by-trial fluctuations of subjective value in monkeys. The authors previously reported the behavioral results of the monkeys trained in the Becker-DeGroot-Marschak (BDM) auction-like task, which allowed them to estimate the monkey's subjective value of liquid rewards on a trial-by-trial basis (Al-Mohammed & Schultz 2022). Building on this work, in the present study, the authors recorded the activity of midbrain dopamine neurons while the monkeys performed the BDM task. The authors show that cue-evoked responses of a subset of dopamine neurons co-fluctuated with the monkeys' subjective values estimated based on the monkey's BDM bids. Furthermore, the activity of a few tens of dopamine neurons was able to predict the monkeys' BDM bids on a trial-by-trial basis.

While there have been a number of studies relating the activity of dopamine neurons and subjective values, these studies often relied on the average activity comparing across different task/trial conditions associated with specific outcome. Using the BDM task, this study demonstrates that the activity of dopamine neurons correlates with subjective values on a trial-by-trial basis above and beyond aggregate estimates over many trials.

The results are important, and I believe that this study warrants publication in Nature Communications. However, I found the authors' framing a little confusing. The distinction between "internal subjective value" and "subjectively weighted objective reward amounts" is unclear and misleading. It is true that the BDM task allows an estimation of subjective value without assuming a function that scales objective reward amounts to subjective value. However, it is possible that subjective values estimated by the BDM method still reflect some scaling of an objective value. It is misleading to suggest that the previous methods using utility functions or temporal discounting did not arrive at "an agent's private internal value" as the individual variability in subjective values can be captured by the variability in utility or discounting functions (e.g. Kable and Glimcher, 2007; PMID: 17982449). The novelty of the present study lies in its ability to infer trial-by-trial fluctuations in subjective values, not in that this study is the first to characterize "an agent's private internal value". As the reviewer 1 pointed out, there have been multiple studies that demonstrated the relationship between the activity of dopamine neurons and subjective values. Historically, the notion that the activity of dopamine neurons mirrors subjective values has been refined gradually through collective efforts. In addition to the papers cited by the authors, Bayer and Glimcher (1995; PMID: 15996553) showed that dopaminergic reward prediction error signals reflect internal estimates of reward expectation on a trial-by-trial basis although this study did not distinguish subjective versus objective values. The demonstration that dopaminergic responses depend on the subject's internal states such as hunger (e.g. Papageorgiou et al., 2016; PMID: 27050518) is also

relevant. It was also shown that dopamine cue responses co-fluctuate with Pavlovian behavioral responses to rewards and punishments on a trial-by-trial basis (Matsumoto et al., 2016; PMID: 27760002). Furthermore, dopamine responses covary with trial-by-trial fluctuations in expected values, belief, or confidence (Lak et al., 2017; Starkweather et al., 2017; Tsutsui-Kimura et al., 2020; PMID: 28285994, 28263301, 33345774). The examination of whether dopamine signals reflect subjective values is, thus, not all-or-none, but builds on efforts of many studies. There have also been many studies studying dopaminergic responses in temporal discounting tasks after Kobayashi and Schultz (2008). It is important to convey a more holistic historical perspective.

The reviewer 2 raised important concerns regarding the novelty and the validity of the study. As I mentioned above, I agree with the reviewer 2 that the distinction between “internal subjective value” and “subjectively weighted objective reward amounts” is very unclear and confusing. However, I believe that demonstrating trial-by-trial co-fluctuations between dopamine responses and subjective values estimated using the BDM bids is highly significant over previous work that largely depended on average responses. The reviewer 2 questioned whether the authors were able to obtain subjective values using the BDM task. I agree with the difficulty in doing so but the authors have done careful characterizations in the previous paper (Al-Mohammed & Schultz 2022). Given these results, the authors have done an impressive job, and the readers will be given enough information to judge the strength of the authors’ claims. In total, I believe that this work contains novel findings that warrant publication in Nature Communications. Nonetheless, it would be useful to tone down the authors’ framing based on economics and to clarify the novelty based on trial-by-trial correlations.

I would suggest the authors to revise the manuscript with the following points in mind:

1. The authors should avoid confusing terminologies such as “internal subjective value” versus “subjectively weighted objective reward amounts”. The novelty of this study lies on their demonstration that dopamine responses co-fluctuate with trial-by-trial estimates of subjective values.

2. The authors should discuss previous studies in a more open-minded manner, reflecting the gradual nature of the studies addressing dopaminergic encoding of subjective values or its trial-by-trial co-fluctuations.

Reviewer #4 (Remarks on code availability):

There is no plan to post their codes on a repository. It would be useful if the data and codes are made open.

Replies to comments by reviewer 4

REVIEWER 4:

This study by Hill and colleagues examined whether the activity of dopamine neurons correlates with trial-by-trial fluctuations of subjective value in monkeys. The authors previously reported the behavioral results of the monkeys trained in the Becker-DeGroot-Marschak (BDM) auction-like task, which allowed them to estimate the monkey's subjective value of liquid rewards on a trial-by-trial basis (Al-Mohammed & Schultz 2022). Building on this work, in the present study, the authors recorded the activity of midbrain dopamine neurons while the monkeys performed the BDM task. The authors show that cue-evoked responses of a subset of dopamine neurons co-fluctuated with the monkeys' subjective values estimated based on the monkey's BDM bids. Furthermore, the activity of a few tens of dopamine neurons was able to predict the monkeys' BDM bids on a trial-by-trial basis.

While there have been a number of studies relating the activity of dopamine neurons and subjective values, these studies often relied on the average activity comparing across different task/trial conditions associated with specific outcome. Using the BDM task, this study demonstrates that the activity of dopamine neurons correlates with subjective values on a trial-by-trial basis above and beyond aggregate estimates over many trials.

The results are important, and I believe that this study warrants publication in Nature Communications. However, I found the authors' framing a little confusing. The distinction between "internal subjective value" and "subjectively weighted objective reward amounts" is unclear and misleading. It is true that the BDM task allows an estimation of subjective value without assuming a function that scales objective reward amounts to subjective value. However, it is possible that subjective values estimated by the BDM method still reflect some scaling of an objective value. It is misleading to suggest that the previous methods using utility functions or temporal discounting did not arrive at "an agent's private internal value" as the individual variability in subjective values can be captured by the variability in utility or discounting functions (e.g. Kable and Glimcher, 2007; PMID: 17982449). The novelty of the present study lies in its ability to infer trial-by-trial fluctuations in subjective values, not in that this study is the first to characterize "an agent's private internal value". As the reviewer 1 pointed out, there have been multiple studies that demonstrated the relationship between the activity of dopamine neurons and subjective values. Historically, the notion that the activity of dopamine neurons mirrors subjective values has been refined gradually through collective efforts. In addition to the papers cited by the authors, Bayer and Glimcher (1995; PMID: 15996553) showed that dopaminergic reward prediction error signals reflect internal estimates of reward expectation on a trial-by-trial basis although this study did not distinguish subjective versus objective values. The demonstration that dopaminergic responses depend on the subject's internal states such as hunger (e.g. Papageorgiou et al., 2016; PMID: 27050518) is also relevant. It was also shown that dopamine cue responses co-fluctuate with Pavlovian behavioral responses to rewards and punishments on a trial-by-trial basis (Matsumoto et al., 2016; PMID: 27760002). Furthermore, dopamine responses covary with trial-by-trial fluctuations in expected values, belief, or confidence (Lak et al., 2017; Starkweather et al., 2017; Tsutsui-Kimura et al., 2020; PMID: 28285994, 28263301, 33345774). The examination of whether dopamine signals reflect subjective values is, thus, not all-or-none, but builds on efforts of many studies. There have also been many studies studying dopaminergic responses in temporal discounting tasks after

Kobayashi and Schultz (2008). It is important to convey a more holistic historical perspective.

The reviewer 2 raised important concerns regarding the novelty and the validity of the study. As I mentioned above, I agree with the reviewer 2 that the distinction between “internal subjective value” and “subjectively weighted objective reward amounts” is very unclear and confusing. However, I believe that demonstrating trial-by-trial co-fluctuations between dopamine responses and subjective values estimated using the BDM bids is highly significant over previous work that largely depended on average responses. The reviewer 2 questioned whether the authors were able to obtain subjective values using the BDM task. I agree with the difficulty in doing so but the authors have done careful characterizations in the previous paper (Al-Mohammed & Schultz 2022). Given these results, the authors have done an impressive job, and the readers will be given enough information to judge the strength of the authors’ claims. In total, I believe that this work contains novel findings that warrant publication in Nature Communications. Nonetheless, it would be useful to tone down the authors’ framing based on economics and to clarify the novelty based on trial-by-trial correlations.

I would suggest the authors to revise the manuscript with the following points in mind:

AUTHORS’ REPLY: Thank you very much for these supportive and constructive comments. We have grouped our replies according to the two specifically proposed changes below.

REVIEWER: 1. The authors should avoid confusing terminologies such as “internal subjective value” versus “subjectively weighted objective reward amounts”. The novelty of this study lies on their demonstration that dopamine responses co-fluctuate with trial-by-trial estimates of subjective values.

AUTHORS’ REPLY: We have removed the confusing terminology and potentially misleading phrasing and now only mention subjective value and its trial-by-trial variation, including in the revised title and throughout the paper. These changes are marked in gray.

REVIEWER: 2. The authors should discuss previous studies in a more open-minded manner, reflecting the gradual nature of the studies addressing dopaminergic encoding of subjective values or its trial-by-trial co-fluctuations.

AUTHORS’ REPLY: Thank you for that conceptually helpful suggestion and the references. We have addressed the issue in a new paragraph in the Introduction. These changes are marked in cyan

REVIEWER: (Remarks on code availability):

There is no plan to post their codes on a repository. It would be useful if the data and codes are made open.

AUTHORS’ REPLY: We have made the data and code publicly available via Figshare. This is now reflected in the section, “Data and code availability”. This change is marked in magenta